# The Interactions among Isolates of *Lactiplantibacillus plantarum* and Dairy Yeast Contaminants: Towards Biocontrol Applications

**Miloslava Kavková** [1,2,*], **Jaromír Cihlář** [1], **Vladimír Dráb** [1], **Olga Bazalová** [1] **and Zuzana Dlouhá** [1,2]

1 Department of Cheese Technologies, Dairy Research Institute, Ltd., Ke Dvoru 12 a, 16000 Praha, Czech Republic; j.cihlar@vum-tabor.cz (J.C.); v.drab@vum-tabor.cz (V.D.); o.bazalova@vum-tabor.cz (O.B.); z.dlouha@vum-tabor.cz (Z.D.)
2 Milcom, Ltd., Culture Collection of Dairy Microorganisms, Ke Dvoru 12 a, 16000 Praha, Czech Republic
* Correspondence: m.kavkova@vum-tabor.cz; Tel.: +420-23-679-012

**Abstract:** Yeast diversity in the cheese manufacturing process and in the cheeses themselves includes indispensable species for the production of specific cheeses and undesired species that cause cheese defects and spoilage. The control of yeast contaminants is problematic due to limitations in sanitation methods and chemicals used in the food industry. The utilisation of lactic acid bacteria and their antifungal products is intensively studied. *Lactiplantibacillus plantarum* is one of the most frequently studied species producing a wide spectrum of bioactive by-products. In the present study, twenty strains of *L. plantarum* from four sources were tested against 25 species of yeast isolated from cheeses, brines, and dairy environments. The functional traits of *L. plantarum* strains, such as the presence of class 2a bacteriocin and chitinase genes and in vitro production of organic acids, were evaluated. The extracellular production of bioactive peptides and proteins was tested using proteomic methods. Antifungal activity against yeast was screened using in vitro tests. Testing of antifungal activity on artificial media and reconstituted milk showed significant variability within the strains of *L. plantarum* and its group of origin. Strains from sourdoughs (CCDM 3018, K19-3) and raw cheese (L12, L24, L32) strongly inhibited the highest number of yeast strains on medium with reconstituted milk. These strains showed a consistent spectrum of genes belonging to class 2a bacteriocins, the gene of chitinase and its extracellular product 9 LACO Chitin-binding protein. Strain CCDM 3018 with the spectrum of class 2a bacteriocin gene, chitinase and significant production of lactic acid in all media performed significant antifungal effects in artificial and reconstituted milk-based media.

**Keywords:** functional traits; antifungal activity; spoilage yeast; *Lactiplantibacillus plantarum*

## 1. Introduction

Lactic acid bacteria (LAB) have been utilised in food and feed processing for centuries [1]. LAB are used as starter cultures for the fermentation of milk, meat, vegetables, and fruits and as adjunct cultures to improve the sensory and health properties of foods. At present, LAB with antimicrobial and antifungal effects are frequently applied as protective agents to limit the use of chemical preservatives and antibiotics, attain green labels, and prolong the shelf life of milk and bakery products [2,3]. *Lactiplantibacillus plantarum*, as representative LAB, is one of the most intensively studied species for its functional properties, which can enhance the quality, safety, and health benefits of foods [1,4,5]. The production of specific organic compounds (organic acids, fatty acids, hydrogen peroxide, bioactive peptides, and proteins) and mutual synergistic effects are strain-specific traits related to the antifungal effect [2,4]. The antifungal effect of *L. plantarum* is supposed to be associated also with the production of bacteriocins [1,4,6] that are primarily related to antimicrobial activity against bacterial contaminants [7,8]. The characterisation of chitinolytic enzymes and its genes [3,9,10] serves other possible products explaining the antifungal activity of

lactobacilli. The genetic background and the resulting metabolic and proteomic expression of these traits are objects of current research [1]. The antifungal effect of *L. plantarum* is frequently verified against filamentous fungi that contaminate dairy and bakery products [3,5–7]. However, fewer studies have examined the antifungal activity of *L. plantarum* against yeast contaminants.

Yeasts are widespread eucaryotic organisms used in the dairy industry as starter and adjunct cultures to manufacture fermented drinks and specific cheeses. Yeasts are responsible for the fermentation process, ripening, and sensory properties of cheeses [11,12]. However, contamination with undesired yeast species remains a problem that affects the quality and safety of dairy products [13]. Detrimental yeast can be introduced from the air, raw material, and dairy equipment. Although hazards analysis and critical control points (HACCP) system is an obligatory hygienic control standard in dairies, the total elimination of yeast in dairy plants is impossible because of their ability to form biofilms in inaccessible technological components, such as tanks and pipes. Yeasts are also present in salt baths, salt brines, and whey [12,13]. The resistance of yeast to sanitation chemicals is also an emerging problem [14]. In well-established, controlled, and stable dairy operations, selected yeast populations are established, which are adapted to specific technology and sanitary conditions. These yeast populations are usually part of the dairy microbial environment, and they do not cause any defects in cheeses. However, the yeast population can increase and transform into undesirable contaminants when the environment or technology is disturbed due to insufficient sanitation or/and technological changes. Besides problems associated with chemical elimination and yeast resistance in dairy, the market demand for clean label products requires the testing and utilisation of bioprotective agent generally recognized as safe (GRAS) in Europe's qualified presumption as safety list (QPS) [15], including *Lactiplantibacillus plantarum*. The application of verified *L. plantarum* strains with functional properties, including antifungal activity against yeast spoilers, is the desired step approaching the sustainable production of fermented dairy products [16,17].

In this work, 20 strains of *L. plantarum* of various origins were screened for genes coding class 2a bacteriocins and chitinase. These strains were tested against 25 yeast strains representing contaminants in cheeses and cheese manufacturing facilities. The extracellular expression of peptides and proteins was verified by using Tricine–SDS electrophoresis. Extracellular peptides and proteins were identified in 30 kDa cell-free filtrates by mass chromatography. The ability to change the pH of the substrate and the production of lactic and acetic acids were measured. The antifungal effects of twenty strains of *L. plantarum* were screened against 25 strains of yeasts isolated from contaminated cheeses and cheese manufacturing facilities. Interactions among yeast and *L. plantarum* strains were first tested by in vitro assays on artificial media [18]. Modified assays with reconstituted milk were used to simulate the matrix and conditions of dairy products [12,19]. The genetic background of bioactive peptides associated with *L. plantarum* and the ability to produce organic acids were used as covariate factors to assess the antifungal effect against yeast. Principal component analysis (PCA) was used to create scatter plots of the strains of *L. plantarum* based on their characteristics.

The aims of this study were to demonstrate that the variability in the antifungal effect of *L. plantarum* strains is related to the origin of the strain and functional traits driven by genes and their phenotypic expression. The study was aimed at specific yeast contaminants belonging to *Candida* spp., *Trichosporon* spp., *Debaryomyces* spp., *Kluyveromyces* spp., and *Geotrichum* sp., obtained from dairy products.

## 2. Materials and Methods

### 2.1. The Functional Traits of L. plantarum Strains

#### 2.1.1. Physicochemical Parameters

The physicochemical parameters were measured for cultures of *L. plantarum* strains cultured in three media: MRS broth (Merck KGaA, Darmstadt, Germany), MRS supplemented with fructose (Lachner, Neratovice, Czech Republic) glucose (Lachner, Neratovice,

Czech Republic), sodium gluconate (Sigma Aldrich, Steinheim, Germany) and maltose (Glentham, Corsham, UK) (MRS-FGGM), and 10% reconstituted milk (RSM) at 30 °C within 24 h. The pH values of the three cultivation media were measured in triplicate using the inoLab pH730 (WTW, Germany) pH meter. Lower fatty acids were detected directly from the three media (MRS, MRS enriched with sugars, and reconstituted milk) by an isotachophoretic analyser (EA 02), VILLA Labeco, Nitra, Slovakia). The results are presented as mg·100 mL$^{-1}$.

### 2.1.2. PCR Amplification of Class 2a Bacteriocin Genes

Class 2a bacteriocin genes were amplified using different primer pairs adopted from the method of a prior study of Wieckowicz et al., 2011 [20]. PCR was performed in a 25 µL reaction volume (0.5 µL of each primer (10 µM), 1 µL of DNA template, 12.5 µL of 2× PPP Taq MasterMix (TopBio, Prague, Czech Republic), and 10.5 µL of ddH$_2$O). The amplification conditions were as follows: pre-heating at 95 °C for 2 min, 35 cycles each of denaturation at 95 °C for 30 s, annealing for 30 s at 44 °C, and extension at 72 °C for 60 s, followed by a final extension at 72 °C for 8 min. PCR products were separated and visualised according to the method described above. PCR products were separated on a GelRed®-stained (Biotium, Fremont, QC, Canada) 2% SeaKem® LE agarose gel (SeaKem® LE; Lonza, Rockland, ME, USA) at 60 V for 120 min and viewed using the Gene Genius Bio Imaging System (Syngene, Frederick, MD, USA).

### 2.1.3. PCR Amplification of the Chitinase Gene

Two different primer pairs were used to detect the presence of the chitinase (chiA) gene, namely, chiAF (5′-ACCCTTCCCACTTTCAAGCC-3′), chiAR (5′-ATATGAGCGTCAGCTCC TCC-3′), chiFEMSF (5′-GATATCGACTGGGAGTTCCC-3′), and chiFEMSR (5′-CATAGAAGT CGTAGGTCATC-3′) according to method used in a previous study [9]. PCR was performed in a 25 µL reaction volume (0.5 µL of each primer (10 µM), 1 µL of DNA template, 12.5 µL of 2× PPP Taq MasterMix (TopBio, Prague, Czech Republic), and 10.5 µL of ddH$_2$O). The amplification conditions were as follows: pre-heating at 95 °C for 2 min, 35 cycles each of denaturation at 95 °C for 30 s, annealing for 30 s at 52/46 °C, and extension at 72 °C for 60 s, followed by a final extension at 72 °C for 8 min. PCR products were separated on a GelRed®-stained (Biotium, Fremont, QC, Canada) 2% SeaKem® LE agarose gel (SeaKem® LE; Lonza, Rockland, ME, USA) at 60 V for 120 min and viewed using the GENE GENIUS Bio Imaging System (Syngene, Frederick, MD, USA).

### 2.1.4. Phenotypic Expression of Bioactive Compounds—Proteomics and Extracellular Production of Bioactive Compounds

Cell-free supernatants were obtained from strains of *L. plantarum* (Table 1).

Cultures grown for 24 h were centrifuged for 15 min (11.000 rpm). The supernatants were filtered through a microbial filter (0.20 µL) (Chromafil CA/S, Macherey-Nagel, Germany). According to the instruction manual, the Amicon ProAffinity Concentrator 30.000 NMWL (Merck Millipore Ltd., Tullagreen, Ireland) was used to obtain the optimal size (kDa) and concentration of extracellular proteins and peptides. Proteins and peptides were isolated from concentrate using ProteoExtract Protein Precipitation Kit (Merck KGaA, Darmstadt, Germany). The obtained pellets were dried and resuspended in 20 µL of Tricine sample buffer (BIO-RAD, Hercules, CA, USA) supplemented with 2-mercaptoethanol (BIO-RAD, USA) (2% *v/v*) and heated at 70 °C for 10 min. Resolving (15%) and stacking gels (4%) and buffers were prepared according to a previous study [21]. Precision Plus Protein™DualXtra Standards (BIO-RAD, Hercules, CA, USA) and Polypeptide SDS-PAGE Molecular Weight Standards™ (BIO-RAD, Hercules, CA, USA) were loaded in wells together with samples (8 µL) under a constant voltage of 125 V until the dye front touched the bottom. The Mini-Protean Tetra Cell Apparatus for 1-D vertical electrophoresis (BIO-RAD) was used. The gels were washed and fixed in a fixing solution (50:10:40/methanol–acetic acid–H$_2$O) for 25–30 min and repeatedly washed in deionised water. The products were

stained by GelCode™Blue Safe Protein (Thermo Scientific, Rockford, IL, USA) in a rocking bath for 1 h. The gels were destained in 10% acetic acid solution. The peptides and proteins were determined directly from the concentrates with 30.000 NMWL on TimsTOF Pro (Bruker Daltonik, Bremen, Germany) with Ultra-High-Performance Liquid Chromatography UltiMate 3000 nano UHPLC (Thermo Fisher Scientific, Waltham, MA, USA) in the Faculty of Science, Dpt. of Chemistry, University of South Bohemia, Czech Republic. MaxQuant software (Max Plankt Institute of Biochemistry, Martinsried, Germany) and the UniProt database were used for data analysis.

The chitinolytic activity of *L. plantarum* strains was tested on basic chitin medium (BCM) (modified method [22]) after 10 days of cultivation at 30 °C. Instead of the original spot-on lawn method, the agar well diffusion method was used. The hydrolysation of chitin by strains of *L. plantarum* was identified as zones around wells in the presence of *L. plantarum* suspension.

**Table 1.** The list of strains of *Lactiplantibacillus plantarum* from the CCDM collection, including their origins and accession numbers.

| *L. plantarum* | CCDM | Origin | GB Accession Number |
|---|---|---|---|
| K19-1 | 3030 | Sourdough | OK189688 |
| K19-2 | 3031 | Sourdough | OK189689 |
| K19-3 | 3048 | Sourdough | OK189690 |
| K20-4 | 3049 | Sourdough | OK189691 |
| CCDM3018 | 3018 | Sourdough | OK189687 |
| CCDM 182 | 182 | Silage | OK189677 |
| CCDM 185 | 185 | Silage | OK189678 |
| CCDM 187 | 187 | Silage | OK189679 |
| CCDM 191 | 191 | Silage | OK189680 |
| CCDM 196 | 196 | Silage | OK189681 |
| L12 | 1091 | Livanjski cheese | MG825683 |
| L16 | 1092 | Livanjski cheese | MG825698 |
| L17 | 434 | Livanjski cheese | MG825687 |
| 21 L24 | 1094 | Livanjski cheese | MG825701 |
| L32 | 1095 | Livanjski cheese | MG825720 |
| CCDM 381 | 381 | Non-pasteurised milk | OK189682 |
| CCDM 383 | 383 | Non-pasteurised milk | OK189683 |
| CCDM 384 | 384 | Non-pasteurised milk | OK189684 |
| CCDM 387 | 387 | Non-pasteurised milk | OK189685 |
| CCDM 391 | 391 | Non-pasteurised milk | OK189686 |
| ATCC14917 | ATCC14917 | Reference strain | [20,23] |

## 2.2. Antifungal Activity Assays

### 2.2.1. *Lactiplantibacillus plantarum* Strains and Cultures

Twenty strains of *Lactiplantibacillus plantarum* isolated from silages, artisanal cheeses, milk, and sourdoughs were obtained from the CCDM collection (CCDM®, Milcom Ltd., Prague, Czech Republic) (Table 1). The reference strain ATCC14917 was used to compare bacteriocins profiles with other strains *L. plantarum* based on the original primer design and other comparative studies [20,23].

All strains were previously identified by sequencing the 16 S rRNA gene. The 16 S rRNA gene was amplified using fD1 (5′-ccg aat tcg aca acA GAG TTT GAT CCT GGC TCA

G-3′) and rP2 (5′-ccc ggg atc caa gct tAC GGC TAC CTT GTT ACG ACT T-3′) primers [24]. The PCR products were treated with 2 μL of ExoSAP-IT[TM] according to the manufacturer's reference manual (ThermoFisher Scientific, Baltic UAB, Vilnius) and sequenced by Eurofins Genomics Germany GmbH (Ebersberg, Germany). Cultures of *L. plantarum* were deposited in the CCDM® collection as deep-frozen (−70 °C) and lyophilised samples.

Lyophilised cultures were revitalised in 16% reconstituted milk at 30 °C for 24 h firstly. Then, *L. plantarum* strains were inoculated (1% *v/v*) in MRS broth (MERCK, Darmstadt, Germany) and cultivated at 30 °C. MRS + FGGM enriched with sugars (Fructose, Glucose, sodium Gluconate, Maltose) according to the M638 recipe (DSMZ, German Collection of Microorganisms and Cell Cultures GmbH, Germany) was used to increase the production of bioactive proteins [25].

### 2.2.2. Yeast Strains and Cultures

Yeast contaminants comprised 25 strains previously isolated from dairy matrices and environments (Table 2). The contaminants were provided by the Culture Collection of Dairy and Bakery Contaminants (CCDBC) (Milcom Ltd., Prague, Czech Republic). The taxonomic affiliation was confirmed by sequencing the ITS region rDNA (ITS1, 5.8 S rDNA, and ITS2) [26,27] by identifying ITS sequences using BLAST alignment tools. The phenotypic characteristics were defined according to a previous study [28]. The strains were deposited on agar slants with supplements. The yeasts were cultivated in yeast malt broth [29] for experimental purposes at 25 °C for 24 h.

**Table 2.** The list of spoilage yeasts in dairy products used in experiments including GenBank Accession numbers and the references.

| Genera | Species | Acronym | Origin | GenBank Accession Numbers | References |
|---|---|---|---|---|---|
| *Candida* | *krusei* | CCDBC 600 | Smeared cheese | OL687493 | |
| | *inconspicua* | CCDBC 601 | Salt bath | OL687494 | |
| | *parapsilosis* | CCDBC 612 | Cheese contaminant | OL687495 | |
| | *intermedia* | CCDBC 622 | Cheese contaminant | OL687496 | |
| | *atlantica* | CCDBC 611 | Salt bath | OL687497 | |
| | *apicola* | CCDBC 610 | Salt bath | OL687498 | |
| | *zeylanoides* | CCDBC 623 | Vegetarian cheese | OL687499 | |
| *Trichosporon* | *domesticum* | CCDM 1062 | Cheese | OL687500 | |
| | *coremiiforme* | CCDBC 607 | Salt bath | OL687501 | |
| | | CCDBC 608 | Salt bath | OL687502 | |
| | *asahii* | CCDBC 624 | Salt bath | OL687503 | [10,11,14,30–32] |
| *Debaryomyces* | *subglobosus* | CCDM 2027 | Traditional Stayer cheese | OL687504 | |
| | *hansenii* | CCDM742 | Cheese | OL687505 | |
| | | CCDBC 615 | Dairy wastewater | OL687506 | |
| | | CCDBC 47 | Kefir | OL687507 | |
| *Kluyveromyces* | *lactis* | CCDM 1054 | Milk | OL687508 | |
| | | CCDBC 617 | Whey | OL687509 | |
| | *marxianus* | CCDBC 620 | Whey | OL687510 | |
| | | CCDM 270 | Milk | OL687511 | |
| | | CCDM 258 | Milk | OL687512 | |
| *Geotrichum* | *candidum* | CCDM 878 | Camembert | OL687513 | |

**Table 2.** *Cont.*

| Genera | Species | Acronym | Origin | GenBank Accession Numbers | References |
|--------|---------|---------|--------|---------------------------|------------|
| | | CCDM 870 | Smeared cheese | OL687514 | |
| *Galactomyces* | *candidum* | CCDM 832 D | Smeared cheese | OL687515 | |
| | *candidum* | CCDM 1053 | Cheese | OL687516 | |
| | | CCDM 1061 | Kefir | OL687517 | |

### 2.2.3. The Overlay Method

*L. plantarum* strains cultivated at 30 °C for 24 h in MRS media were used in the modified overlay method [3,13,33]. The *L. plantarum* strains were streaked in two lines on MRS agar and cultivated anaerobically at 30 °C for 48 h. Cooled 0.7% malt extract agar was inoculated with yeast strains at a concentration of $10^4$ CFU in 1 mL and gently poured on MRS medium with *L. plantarum* strains. Plates were cultivated at 25 °C in aerobic conditions. The growth of yeasts was assessed after 12, 24, 48, and 72 h. The final effect of *L. plantarum* strains on the yeast was evaluated according to an index scale of 1–7 (Table 3). Each combination of lactobacilli and yeast strain included three replicates.

**Table 3.** The index scale with a description of scale values from the overlay method on artificial media. Three levels of inhibition represent the level of inhibition for PCA analysis.

| Description | Index Scale | Inhibition |
|-------------|-------------|------------|
| Yeast colonies cover the whole Petri dish | 1 | No inhibition |
| Yeast colonies are sparse between *L. plantarum* colonies compared with the rest of the Petri dish | 2 | Partial inhibition |
| Yeast colonies occur outside *L. plantarum* colonies | 3 | |
| Yeast colonies occur until 5 mm from the outer edges of *L. plantarum* colonies | 4 | |
| Yeast colonies occur up to 5 mm from the outer edges of *L. plantarum* colonies | 5 | |
| Yeast colonies are only on the edges of the Petri dish | 6 | Total inhibition |
| No yeast colonies on the Petri dish | 7 | |

### 2.2.4. The Agar Layer-Diffusion Method

A reconstituted milk was used in the modified agar layer diffusion method [34]. The strains of *L. plantarum* were cultivated overnight in 10% reconstituted milk at 30 °C. Then, the cultures were mixed with malt extract agar cooled to 45 °C; thus, the final concentration of the medium was 30% reconstituted milk with the tested strain of *L. plantarum*. To improve the visualisation of live yeast cells, 10 μL of neutral red solution (3% *v/v*) (Sigma-Aldrich, Darmstadt, Germany) was added to the mixture [35]. Yeast cultures grown for 24 h in yeast malt broth were diluted in Ringer's solution (Sigma-Aldrich, Darmstadt, Germany) to $1 \times 10^5$ CFU in 1 mL. Four drops (5 μL) were placed on solidified media containing 30% reconstituted milk with *L. plantarum* strains. The Petri dishes were incubated at 25 °C. The growth of the yeast colonies was assessed every 12 h and stopped after 72 h. Yeast grown on media with milk without lactobacilli and on MEA (0.7%) (Sigma-Aldrich, Darmstadt, Germany) were used as control variants. Each combination of *L. plantarum* strain and yeast strain was analysed in triplicate.

### 2.3. Statistical Analyses

Data for physicochemical parameters were subjected to statistical analysis using ANOVA with a factorial design in Statistica Software v 12.0 (StatSoft, Tulsa, OK, USA).

Significant differences among the tested groups were determined using Tukey's HSD test at significance $p \leq 0.05$. Variability in the antifungal effect based on index scale values was analysed by ANOVA with a factorial design, Statistica Software v 12.0 (StatSoft Europe, Hamburg, Germany). Index values were first log-transformed and presented as the mean with standard error of the means (s.e.m.) because of discrepancy of sample's means [17,36]. Significant differences among the tested groups were determined using Tukey's HSD test at significance $p \leq 0.05$. Analysis of covariance (ANCOVA) was used to compare the combinatory effect of functional traits of lactobacilli (organic acids, pH, bacteriocin genes, and chitinase genes) on the variability in the antifungal effect. Parameter values were first log-transformed. The significance was tested at $p \leq 0.05$ (post hoc Tukey's HSD Test). The antifungal effects of *L. plantarum* strains on artificial media were divided into three categories: i.e., total inhibition, inhibition, and no inhibition. The effects were first summarised by using principal component analysis (PCA) using their correlation matrix. The functional traits were converted to a log-transformed dataset to summarise the multiplicative changes.

The Hedonic rating scale was used for data scoring and analysed in Statistica Software 12.0 (StatSoft Europe, Hamburg, Germany) using multivariate techniques [23]. The variability in the average colony size (mm) of yeast was the main parameter used to assess the antifungal activity of lactobacilli in the RSM medium. The dataset obtained in this experiment was managed as described above. The average colony size and the zone around the yeast colony were summarised by the PCA model using their correlation matrix and visualised as PCA biplots and PCA scatter plots. The sensitivity of 25 yeast strains to strains of *L. plantarum* with defined traits was analysed by ANCOVA for groups represented by yeast genera to avoid statistical error during non-parametric testing because of unequal N.

## 3. Results and Discussion

First, the functional traits of *Lactiplantibacillus plantarum* were tested. The production of lactic acid, acetic acid, and changes in the substrate pH by strains of *L. plantarum* were evaluated in three tested media. Genomic methods were used to detect the genes responsible for producing bioactive peptides and chitinase.

The presence of extracellular bioactive peptides in cell-free filtrates was verified by using Tricine -SDS-PAGE on 15% tricine gels. The spectra of extracellular peptides in cell-free filtrates were determined by using mass spectrometry. The results obtained from these analyses were used as characteristics of *L. plantarum* strains in statistical models to help us clarify the background of their antifungal activity.

### 3.1. The Functional Traits of L. plantarum Strains

3.1.1. Physicochemical Parameters

The physicochemical parameters of lactobacilli were tested and compared among strains of different origins cultured in the three media types. The production of lactic and acetic acid and the pH of media varied significantly for all tested factors (Table 4), i.e., strains of *L. plantarum*, the group of origin, and substrate.

Production of lactic acid varied significantly among all strains of *L. plantarum* and the groups of origin (Figure A2). The acetic acid production varied significantly within the strains of *L. plantarum*, but there were no evident differences among the origin groups. The variability in lactic acid production is apparent among all the strains depending on the cultivation substrate. The strains originating from milk (CCDM 384, 391) produced the highest amount of lactic acid in MRS and MRS + FGGM. Despite that, these strains made the lowest amount of lactic acid in RSM. The addition of sugars (FGGM) significantly increased lactic acid production by all the strains of *L. plantarum* in MRS. In this medium, the strains from milk and silages were significant producers of lactic acid. In RSM, the highest amount of lactic acid was measured when strains from silage (CCDM 182), cheese wild types (L16, L17) and CCDM 3018 from sourdough were cultivated. The pH value of MRS media (5.57) was reduced by *L. plantarum* strains up to 30% without significant

difference. The significant variability among the strains within the groups of origin was noted in RSM (Table A1, Figure A1). The strains obtained from sourdoughs decreased the original pH value (6.62) of RSM by about 44% at maximum. The strains of *L. plantarum* can utilize substrates rich in starch and sugars to produce lactic acid [37] and other short-chain organic acids. Lactic acid production is supposed to be the base of the antifungal effect [21]. Besides the lactic acid, the production of benzoic acid and diacetyl were confirmed in dairy matrices with *L. casei* [38]. The content of lactic acid in dairy matrix can be increased when supplements such as whey or milk powder are added [39]. The antifungal effect of lactobacilli is described mainly at the species-level under in vitro conditions [19,40]. Thus, the production of organic acids with possible antifungal effects remains optimal in these tests but becomes changed under technological conditions in dairies (salt, temperature, and microbial cultures).

**Table 4.** Physicochemical parameters of media inoculated with *L. plantarum* strains (ANOVA, Factorial design, Post Hoc Tukey's HSD Test, Statistica v.12.0). Factors (F1, F2, F3) are represented by four groups of lactobacilli origin (F1), twenty strains of lactobacilli (F2), both summarised in Table 1, and three types of the cultivation media (F3) (MRS, MRS + FGGM, milk). The asterisk indicates significant differences. * Significant at $\alpha \geq 0.05$.

| Factors Tested | | Lactic Acid | | Acetic Acid | | pH | |
|---|---|---|---|---|---|---|---|
| | df | F | *p* | F | *p* | F | *p* |
| Groups (F1) | 3 | 2.11 | 0.44 * | 2.19 | 0.101 | 3.14 | 0.033 * |
| Strains (F2) | 19 | 41.04 | 0.001 * | 6.3 | 0.001 * | 12.80 | 0.001 * |
| Medium (F3) | 1 | 58.90 | 0.001 * | 230.1 | 0.001 * | 246.36 | 0.001 * |
| F1 × F2 | 6 | 10.75 | 0.001 * | 3.88 | 0.001 * | 14.50 | 0.001 * |
| F1 × F3 | 6 | 3.16 | 0.01 * | 1.03 | 0.42 | 5.21 | 0.001 * |
| F2 × F3 | 6 | 3.43 | 0.006 * | 1.70 | 0.136 | 128.40 | 0.003 * |
| F1 × F2 × F3 | 4 | 8.22 | 0.001 * | 1.59 | 0.079 | 16.30 | 0.001 * |

### 3.1.2. Bacteriocins

Bioactive peptides from *L. plantarum* with antimicrobial and antifungal effects have been described in recent studies [1,20,23]. Most of the defined peptides were also detected when lactobacilli were tested against filamentous fungi [41]. The production of antifungal peptides by *L. rhamnosus* and *L. paracasei* in milk matrices targeted against *Debaryomyces hansenii* was described by McNair et al. [42]. Although many functional properties of *L. plantarum* and other lactobacilli based on the production of bioactive peptides have been described [1,7], the specificity of strains against targeted yeast contaminants is species-specific and variable. The gene coding for the related antifungal peptides is unknown or strain-specific; thus, screening *L. plantarum* strains for such traits is ineffective. As bacteriocins are the best-described bioactive peptides with a broad spectrum of antimicrobial and antifungal effects, the tested strains of *L. plantarum* were screened for their genetic backgrounds. Table 5 summarizes the characteristics of *L. plantarum* strains based on the presence of bacteriocin genes belonging to class 2a.

All *L. plantarum* strains encoded class 2a bacteriocin genes belonging to clade 1 (Figure A3). The profiles of PCR products in individual clades showed that not a single strain of *L. plantarum* has the same pattern as reference strain ATCC 14917. In addition, more than half of our strains encoded bacteriocin from clade 2, excluding strains CCDM 182, 185, 381, 383, 384, 387, 391, and 3018, which did not show amplified products (Figure A4). Clade 4 bacteriocin was successfully amplified using the first primer pair in all strains except CCDM 381 (Figure A5). Amplification with the second primer pair failed in most strains, excluding L12 and L16. L12 yielded a product size comparable to the PCR products in Więckowicz et al, 2011 [20]. Strain L16 produced a band of smaller size than expected (Figure A6). Non-specific primer annealing occurred during the amplification of bacteriocins from clade 5, regardless of the forward primer used (Figures A7 and A8). Variability in

PCR products among lactobacilli was detected after using the first primer pair (Figure A7). Half of the strains yielded PCR products of approximately 50 bp, corresponding to the values presented in previous study [20]. Strains K19-3 and CCDM 3018 produced a band with a size close to that of the expected product. Strains L12-16 and CCDM 381, 383, and 384 only yielded a spectrum of larger non-specific bands. A similar result was observed when using the second primer pair (Figure A8). The DNA of a single strain, CCDM 381, was not amplified using a second primer pair. Single bands of approximately 500 bp and 1000 bp were produced by strains K19-3 and K20-4, respectively. The sizes of both products exceeded the size declared in [20]. Strains K19-1, K19-2, CCDM 191 and 196, and L12-17 produced a band of the appropriate size. The functional diversity of strains *L. plantarum* was confirmed recently by comparative genomic analyses of 54 genome sequences pointed on the biosynthesis of bacteriocins. The three evolutionary lineages of *L. plantarum* varied in the biosynthesis of plantaricin [43]. Thus, strains from individual lineages performed diverse functional traits. The tricine-SDS-PAGE method was used to screen the extracellular expression of bacteriocin genes into the substrate. The extracellular proteins and peptides isolated from cell-free supernatants using AMICON ultrafilters with 30 MWO corresponded to sizes of 25–20 kDa, 20 kDa, 10 kDa, and less than 5 kDa (Figure 1).

**Table 5.** The presence of bacteriocin Class 2a genes in cell-free filtrate with 30.000 NMWL.

| | **Bacteriocins Class 2a** | | | | | |
|---|---|---|---|---|---|---|
| *L. plantarum* | **Clade 1** | **Clade 2** | **Clade 4-1** | **Clade 4-2** | **Clade 5-1** | **Clade 5-2** |
| K19-1 | + | + | + | − | + | + |
| K19-2 | + | + | + | − | + | + |
| K19-3 | + | + | + | − | +50 bp, +150 bp | +(cca 500 bp) |
| K20-4 | + | + | + | − | +50 bp | +(cca 1000 bp) |
| CCDM 3018 | + | − | + | − | +50 bp, +150 bp | multiple bands |
| CCDM 182 | + | − | + | − | + | multiple bands |
| CCDM 185 | + | − | + | − | + | multiple bands |
| CCDM 187 | + | + | + | − | + | multiple bands |
| CCDM 191 | + | + | + | − | + | + |
| CCDM 196 | + | + | + | − | + | + |
| L12 | + | + | + | +150 bp, −50 bp | multiple bands | + |
| L16 | + | + | + | −150 bp, +50 bp | multiple bands | + |
| L17 | + | + | + | − | multiple bands | + |
| L24 | + | + | + | − | multiple bands | multiple bands |
| L32 | + | + | + | − | multiple bands | multiple bands |
| CCDM 381 | + | − | − | − | multiple bands | − |
| CCDM 383 | + | − | + | − | multiple bands | multiple bands |
| CCDM 384 | + | − | + | − | multiple bands | multiple bands |
| CCDM 387 | + | − | + | − | multiple bands | multiple bands |
| CCDM 391 | + | − | + | − | + | multiple bands |
| ATCC14917 | + | + | + | +150 | + | +(cca 500 bp) |

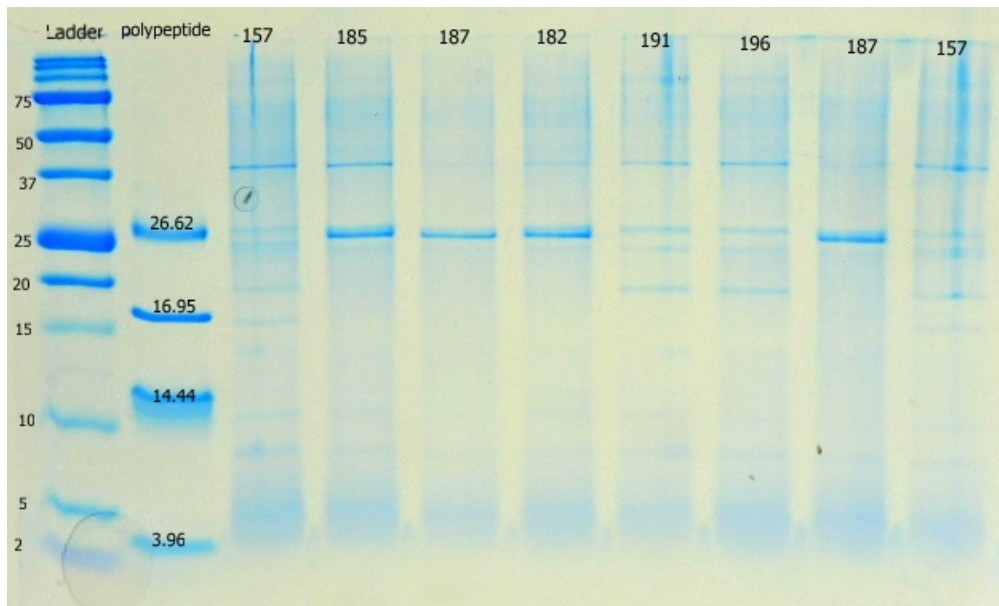

**Figure 1.** The proteomic profile of extracellular products of *L. plantarum* strains isolated from silages on 15% Tricine-SDS-PAGE gel. Chitinase bands in the 20 kDa range are visible in samples 191 and 196. Sample 157 is not included in this study. Precision Plus Protein™ Dual Xtra Standards was used as the peptide standard. Polypeptide Molecular Weight Standards™ represents insulin b chain (3.96 kDa), α-lactalbumin (14.437 kDa), myoglobin (16.95 kDa) and triosephosphate isomerase (26.62 kDa).

Mass spectrometry analysis on filtrates and products from tricine gels showed a broad spectrum of proteins that are metabolic products of *L. plantarum* described in previous studies [7,40,44]. Although the bands less than 5 kDa were evident in extracellular filtrates of *L. plantarum* no known bacteriocin-like or antifungal peptides were identified among these peptides.

### 3.1.3. Chitinase

Currently, there are only a few studies about the detection of chitin-binding proteins (CBPs) and the chitinolytic activity of *L. plantarum* [9,45]. In most of the studied *L. plantarum* strains, the *chiA* gene was successfully amplified using *chiFEMSF* and *chiFEMSR* primers [9] (Table 6, Figure A9), from which it was assumed that these strains encode the *chiA* gene. However, the amplification with the ChiAF and ChiAR primers based on *Lactococcus lactis* subsp. *lactis chiA* sequences [30] failed, possibly because they are not compatible with *L. plantarum* DNA. The *chiA* gene was present in all strains from silages, whereas the lack of *chiA* was noted in most of the strains isolated from milk. The extracellular proteins and peptides isolated from cell-free supernatants using AMICON ultrafilters with 30 MWO corresponded to sizes of 25–20 kDa, 20 kDa, 10 kDa, and less than 5 kDa (Figure 1).

In most of the tested *L. plantarum* strains, chitinase was the only expressed extracellular protein with antifungal activity detected in the products. Although the 9 LACO Chitin-binding protein (22.199 kDa) was identified in most concentrates of cell-free supernatants from *L. plantarum* strains with the *chiA* gene, there were some discrepancies in strains CCDM 387 and L32. In these two strains, the *chiA* gene was not detected, but the 9 LACO Chitin-binding protein was detected in the concentrate and directly from 20 kDa products on 15% tricine gels. The visibility of chitinase on 15% Tricine gels depends on the absolute quantity of the peptide in the sample, i.e., the previous concentration of cells in the medium and conditions for the phenotypic expression of genes [46]. The conditions for the expression of CBP genes in artificial media and milk matrices are still unknown. The number of chitinase gene copies in genomic DNA might differ from strain to strain, so the result depends on the sensitivity of the PCR method. TimsTOF Pro with Ultra-High-Performance Liquid Chromatography UltiMate 3000 nano UHPLC remains a highly sensitive method

to screen peptides or proteins present in small amounts in the concentrate [47,48]. The strains of *L. plantarum* were screened for chitinase activity in vitro using colloidal chitin in BCM medium combined with the agar well diffusion method. Clearing zones have never occurred; instead, the irregular hyaline zones occurred around the wells with suspensions. Seven *L. plantarum* strains (L17, L16, K19-2, 191, 3018, 185, and 187) hydrolysed colloidal chitin. The same strains were tested positively for the presence of chitinase genes. There are no known conditions for chitinolytic activity of CBP nonhydrolytic proteins against the yeast and fungi in milk matrices. Primarily CBP proteins bind to N-acetylglucosamine residues in chitin structures in the fungal cell walls [45]. The research on *Candida albicans* and *G. glabrata* confirmed that the inhibition of candida growth by chitinase represented by hydrolase Msp1 from *L. rhamnosus* GG is supported with lactic acid, particularly L-lactic acid isomer that potentially increases the sensitivity of hyphal surface to chitinase [10].

**Table 6.** The presence of chitinase *chiA* in DNA of tested *L. plantarum* strains. Presence of 9 LACO Chitin-binding protein (22.199 kDa) in cell-free filtrate with 30.000 NMWL.

| | | Chitinase |
|---|---|---|
| *L. plantarum* | *chiA* | **9 LACO Chitin-Binding Protein OS** |
| K19-1 | − | + |
| K19-2 | + | + |
| K19-3 | + | + |
| K20-4 | + | + |
| CCDM 3018 | + | + |
| CCDM 182 | + | + |
| CCDM 185 | + | + |
| CCDM 187 | + | + |
| CCDM 191 | + | + |
| CCDM 196 | + | + |
| L12 | + | + |
| L16 | + | + |
| L17 | + | + |
| L24 | − | − |
| L32 | + | − |
| CCDM 381 | + | + |
| CCDM 383 | − | − |
| CCDM 384 | + | − |
| CCDM 387 | − | + |
| CCDM 391 | − | − |

### 3.2. Antifungal Activity Assays

#### 3.2.1. Antifungal Activity of *Lactiplantibacillus plantarum* Strains

The antifungal activity of twenty strains belonging to *L. plantarum* was tested against 25 strains of yeast contaminants. The overlay method on artificial media represented the optimal in vitro conditions for interacting organisms, lactobacilli, and yeast. In overlay assay, the inhibitory effect against yeast varied significantly among *L. plantarum* strains (Figure 2, Table 7) and yeast strains. The numbers of non-inhibited, partially inhibited, and fully inhibited yeast strains are presented in Figure 2. The results show that the number of fully inhibited yeast strains was highest for *L. plantarum* strains isolated from sourdough (3018 and K19-3), silage (196), and cheese-wild types (L16, L24). The strain L16 inhibited

eight yeast strains which is a maximal number of total inhibitions. The strains originating from raw milk proved rather partial inhibitions against the yeast inhibited and less rate of total inhibition. Strain CCDM 384 weakly suppressed yeast growth on artificial media, reflected by the high number of low-indexed inhibitions (Figure 2). This strain isolated from milk can partially inhibit a broad spectrum of yeast strains at a low level.

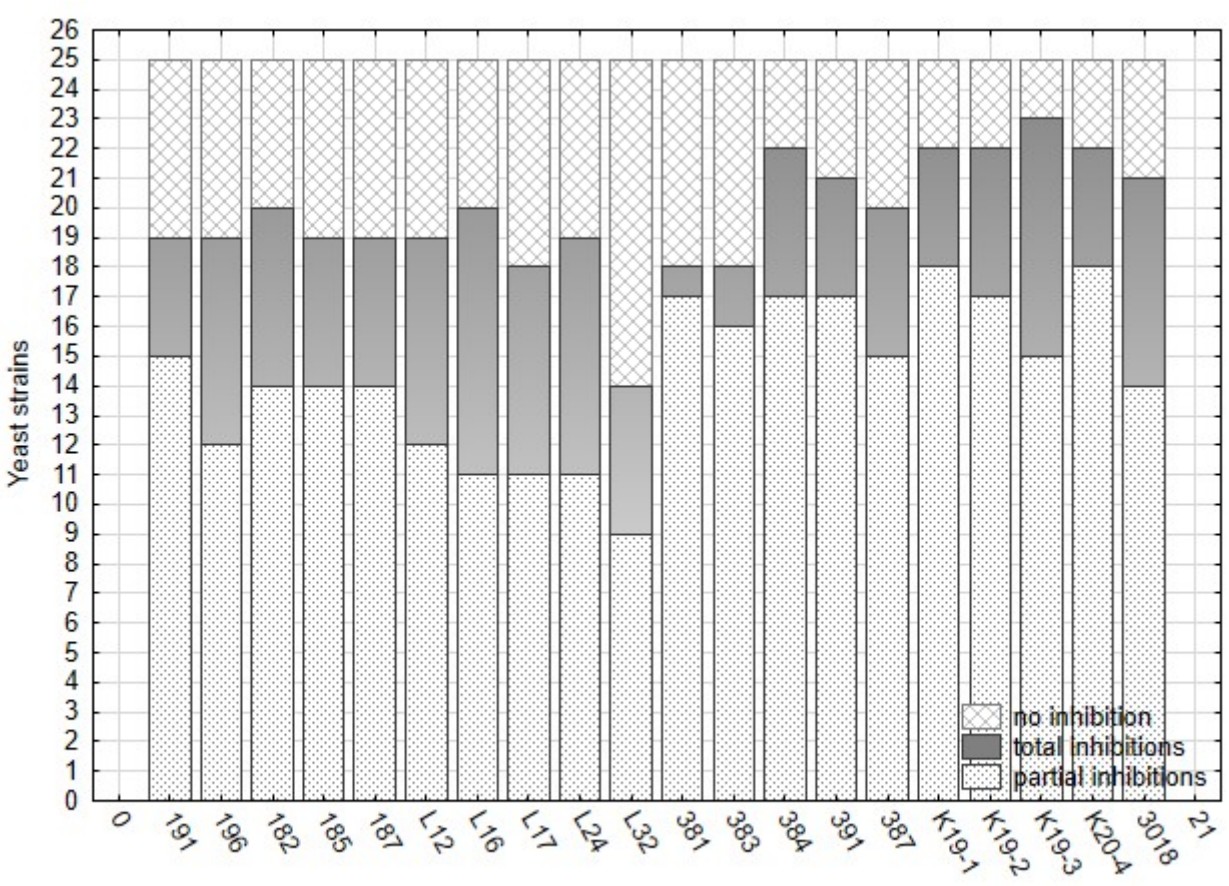

**Figure 2.** The numbers of yeast strains (25) showing no inhibition, total inhibition (6–7), and partial inhibition (2–5) by *L. plantarum* strains (20) based on index scale values from the overlay method (Table 3) on artificial media.

**Table 7.** The interactions between lactobacilli and yeast strains in the in vitro overlay method, including two control variants for each yeast strain. ANOVA, Statistica Software v. 12, Post Hoc Tukey's HSD Test, * Significant at $\alpha \le 0.05$.

| Factor | DF | F | *p* |
|---|---|---|---|
| *L. plantarum* strains (F1) | 20 | 94.5 | 0.001 * |
| *L. plantarum*-groupes of origin (F2) | 4 | 358.6 | 0.001 * |
| Yeast strains (F3) | 24 | 424.5 | 0.001 * |

The overall inhibitory effect of *L. plantarum*, including partial and total inhibition, reached 92% (K19-3) and 88% (384). Although the strains from milk inhibited yeast more potently than those from raw cheese and silages, the inhibitory effect was partial and temporary. The results from the overlay method confirmed that all strains of lactobacilli inhibited all yeast contaminants in general compared with the control variants (Figure 3). The data presented in Figure 3 are supported with detailed statistical parameters included

in Table A2. The complete dataset related to testing 20 strains of *L. plantarum* against 25 strains of the yeast obtained from analyses of variance is supplemented as Table S3.

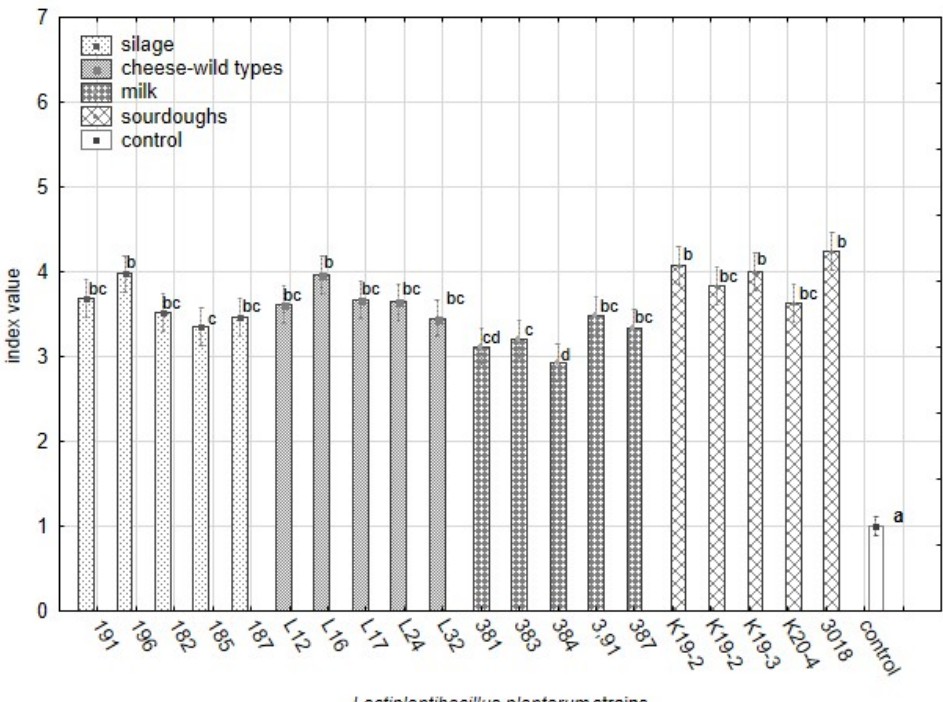

**Figure 3.** The variability in the antifungal effect against 25 yeast strains among *L. plantarum* strains from different origins based on index scale values from the overlay method. The data represent the means and SEM (ANOVA, Statistica v. 12, Post Hoc Tukey's HSD Test). The index letter indicates significant differences. Significant at $\alpha \leq 0.05$.

The intensity of the inhibitory effect was shown to be driven by different functional traits of lactobacilli [47,49]. The functional properties of lactobacilli, such as pH, production of LA and AA, bacteriocin and chitinase genes, were tested as covariates. The dimensional variability and interactions among the traits were evaluated by principal component analyses (PCA) (Figure 4). Partial inhibition (index value 2–5) is distinct from total inhibition (index value 5–6) and no inhibition (index value 1). The first two components explained 99.9% of the variance, with a cophenetic coefficient of 0.97. The first component explained 61.56% of the variability and showed that the inhibitory effect (total inhibition, partial inhibition, or no inhibition) of lactobacilli on yeast is influenced by different functional properties of *L. plantarum* strains. The total inhibition of yeast was positively correlated with the spectrum of genes for bacteriocins class 2a and chitinase. Although the spectrum of genes was shown as the significant trait allied with the antifungal activity, the base of this phenomenon is unclear. The six strains of *L. plantarum* (3018, K19-3, 196, L12, L16, L24) with significant antifungal activity varied in clades 4-1 and 5-1 of bacteriocins class 2a. The role and importance of individual genes for the final antimicrobial and antifungal effects are unknown. The strain L24 lacked the chitinase gene and protein. Partial inhibition was positively correlated with the production of lactic acid and acetic acid. The second component explained 38.44% of the original variability and positively correlated with the pH of the medium and the content of lactic and acetic acids. The distribution of *L. plantarum* strains (Figure 5) according to index values in the factorial space explained the indexed inhibitory effect correlated with functional traits. The six strains of *L. plantarum* that inhibited the most yeast strains are cumulated at the bottom right corner of the scatter plot. The studies aimed at antifungal effect are using evaluating scales representing mostly just the occurrence of inhibition zones. The yeast species representing dairy contaminants remain a group of diverse strains that performed unlike properties such

as adaptability, tolerance, and sensitivity. To evaluate mutual interactions with lactobacilli strains, a detailed evaluating scale respecting both organisms is necessary.

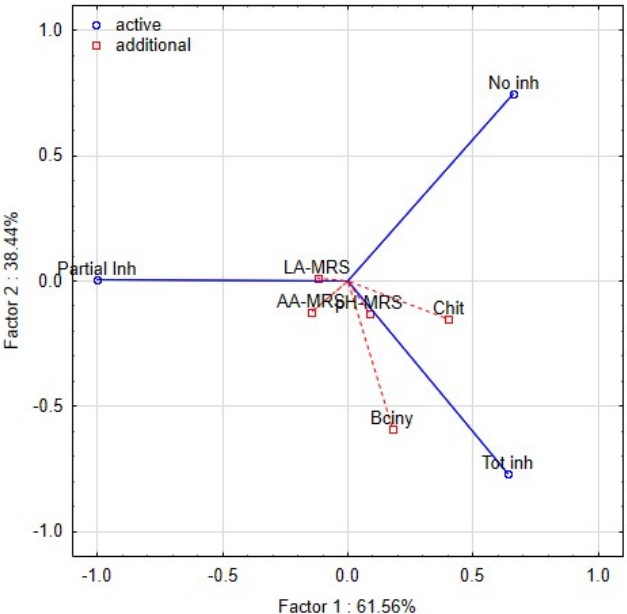

**Figure 4.** Biplot representing the first two components of PCA performed on the standardised values of index logarithms representing the interaction among lactobacilli and yeast contaminants in the overlay method. The functional properties of lactobacilli are additional factors—pH, LA (lactic acid), AA (acetic acid), Bciny (bacteriocins), Chit (chitinase).

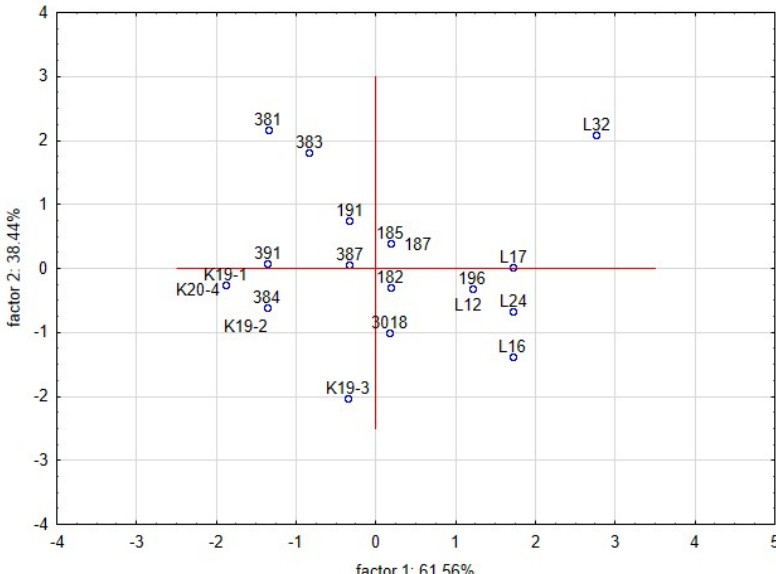

**Figure 5.** Principal component analysis (PCA) on the standardised value of index logarithms representing the interaction among lactobacilli and yeast contaminants in the overlay method. Scatter plot of the position of strains of *L. plantarum* is based on correlations among its functional properties. The interaction between the *L. plantarum* strains and yeast contaminants under the RSM conditions significantly influenced the average size of yeast colonies (mm). All *L. plantarum* strains significantly suppressed the growth of yeast colonies compared with the control (Figure 6, Table 8).

**Table 8.** The factors influencing in the average size of yeast colonies (mm) on RSM medium with *L. plantarum* strains, including the control variant, for each yeast strain. Factorial ANOVA, Statistica Software v. 12, Post Hoc Tukey's HSD Test, * Significant at $\alpha \leq 0.05$.

| Factor | DF | F | *p* |
|---|---|---|---|
| *L. plantarum* strains (F1) | 20 | 94.5 | 0.001 * |
| Groups of origin (F2) | 4 | 424.5 | 0.001 * |
| F1 × F2 | 504 | 5.83 | 0.001 * |

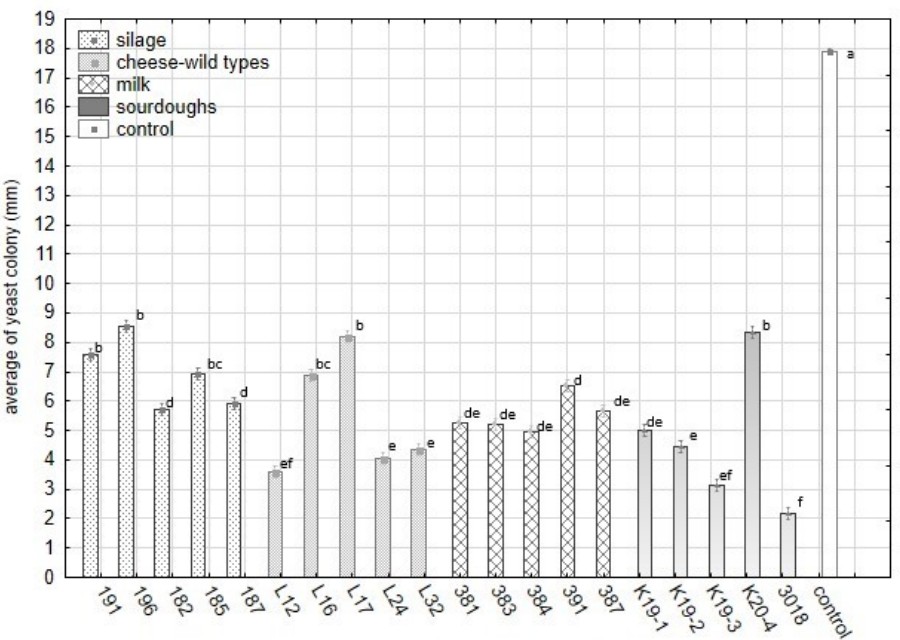

**Figure 6.** The differences in average sizes (mm) of yeast colonies growing on media with reconstituted milk and lactobacilli strains included in groups of origin (Factorial ANOVA, Statistica Software v. 12, Post Hoc Tukey's HSD Test). The bars represent means and s.e.m. The index letter indicates significant differences. Significant at $\alpha \leq 0.05$.

The data shown in Figure 6 are supported by Table A3 including detailed statistical parameters. The complete dataset of analysis of variance based on the average of yeast colonies on media with RSM is supplemented as Table S4. The significance of factors influencing the variability within strains of *L. plantarum* and groups of origin is summarised in Table 8. The strains from sourdoughs (3018, K19-3) and cheese-wild types (L12, L24, L32) suppressed the yeast growth on RSM enriched medium significantly compared with control and others. The spectrum of effective strains from sourdoughs on RSM medium is the same compared to overlay assay on artificial media. The effective strains belonging to cheese-wild types (L12, L24, L32) differed from those effective on synthetic media. Notably, the strain L32 inhibited yeast growth significantly under RSM conditions and showed a weak antifungal effect on artificial media. The strains originating from silages suppressed the yeast growth weakly in RSM.

The PCA analyses were used to relate the yeast growth with characteristics of *L. plantarum* strains in RSM matrix (Figure 7). Inhibitory zones were observed in only a few combinations of lactobacilli and yeast strains. The proteolytic activity of yeast strain can also induce the clearing zone; thus, the comparison with the control variant is necessary. The first two components represented 99.9% variability. The first component explained 61.63% of the variability. The growth of yeast colonies was negatively correlated with the production of organic acids and bioactive proteins, and peptides. The second component

demonstrated 38.37% of the variability and was positively correlated with the pH of RSM. The strains of *L. plantarum* in the scatter plot of PCA (Figure 8) are clustered according to their inhibitory effect against the yeast. The strains in the bottom left corner (L12, L32, L24, K19-3, 3018) represent the most effective strains. They are positively correlated with bacteriocins and organic acids, whereas the strains in the upper left corner are correlated with pH value and the occurrence of the zone. The antifungal effect of strains from sourdoughs, 3018 and K19-3, was the most effective on artificial and RSM-based medium. Besides the bacteriocins and chitinase profile, strain 3018 produced the lactic acid in MRS and RSM at the highest stable amount to compare the rest of the strains. Thus, concerning medical research on *C. albicans* [10], the intensive production of lactic acid can also support chitinase's role in antifungal action.

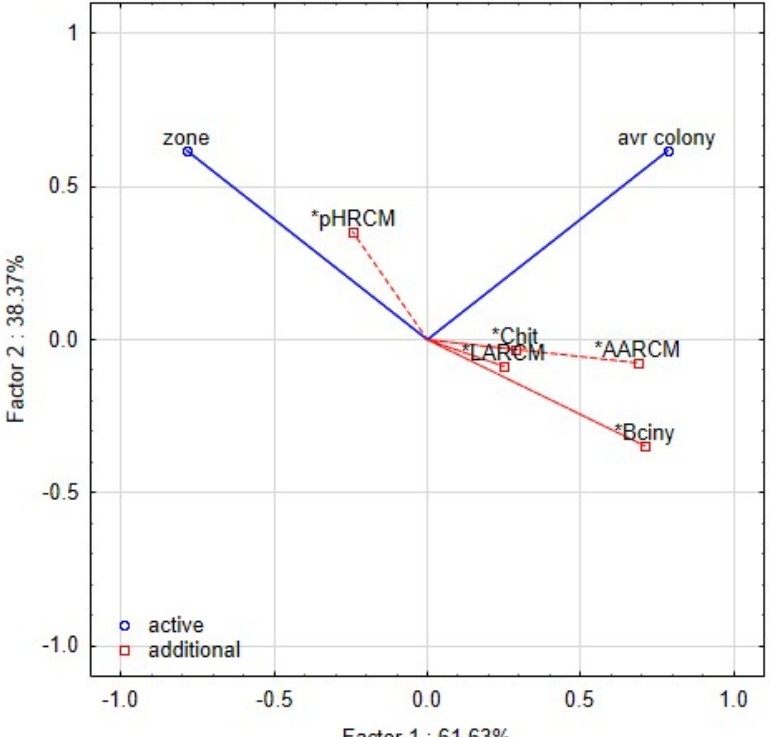

**Figure 7.** Biplot representing the first two components of PCA performed on the standardised values of index logarithms representing the interaction among lactobacilli and yeast contaminants in the overlay method. The functional properties of *L. plantarum* strains are included as additional factors. Asterisk notes the functional traits—pH, LA (lactic acid), AA (acetic acid), Bciny (bacteriocins), Chit (chitinase).

The sources of isolates/strains and strain-associated traits such as bile resistance, adaptability, and the production of extracellular proteins play essential roles in phenotypic characteristics [49]. The strains isolated from sourdoughs had the most substantial suppressive effect against yeast. In our study, the strains of *L. plantarum* originating from sourdoughs inhibited up to 90% of the 25 yeast strains. Most published studies on *L. plantarum* strains producing bioactive compounds and exhibiting antifungal activity originated from sourdoughs and related bakery material [3,4,31]. In addition, wild strains of *L. plantarum* isolated from raw ewe cheeses inhibited yeast growth on both media tested. The functional properties of lactobacilli isolated from raw milk and artisanal cheeses have been characterised in recent studies [7,50]. Strains 191 and 196 from silages exerted effective antifungal activity against yeast in an overlay test on artificial media. The *L. plantarum* strains from silages reduced the yeast growth in RSM medium to a significantly lesser extent than those from raw cheeses and sourdoughs. The properties of lactobacilli isolated from silage were successfully tested to improve animal feed quality [51]. The strains obtained from milk showed different profiles of bioactive

proteins and the weakest antifungal effect among the groups of origin. The combination of the characteristics of *L. plantarum* and the sensitivity of yeast strains generally influenced the results. Potential antifungal compounds (peptides and organic acids) produced by strains of *L. paracasei* and *L. rhamnosus* [42], *L. rhamnosus*, and *L. jensenii* [13,52] were identified in dairy matrices in recent studies [4]. The spectrum of bioactive peptides produced under the RSM condition might differ from that in optimised growing media. The inhibition of yeast growth can be limited by time; i.e., lactobacilli and their products may only postpone development, and/or they can influence the morphological features of yeast, such as the density and size of the yeast colonies. In a recent study, the genomic comparison based on clusters of orthologous groups did not show intraspecific diversity of plant-associated strains belonging to *L. plantarum*, but the variability was significantly supported by phenotypic studies [53]. The inhibitory activity against *Saccharomyces cerevisiae* confirmed significant variability within strains [53] from fermented fruits and grains.

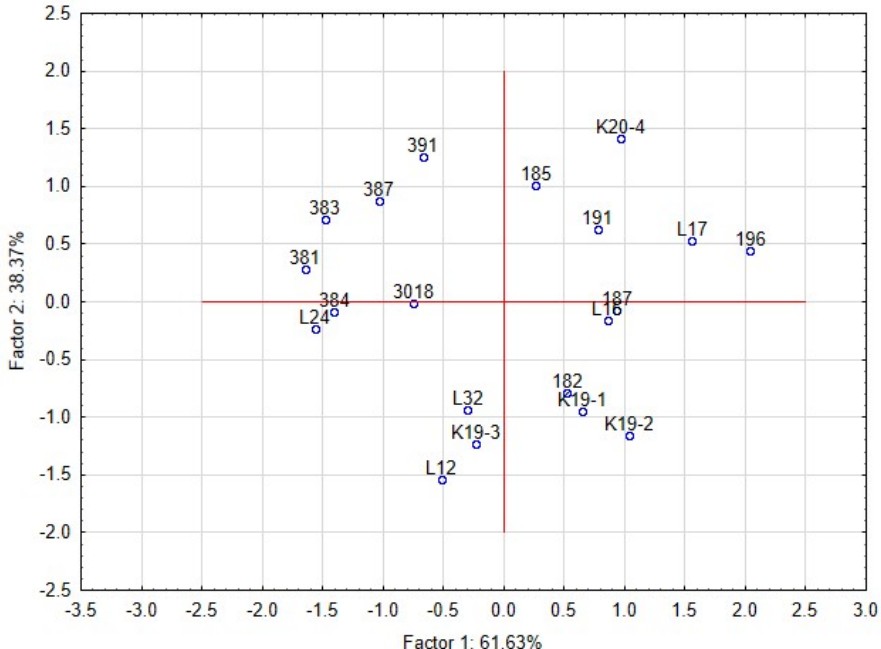

**Figure 8.** Principal component analysis (PCA) on the standardised values of the average size of yeast colonies (mm) representing the interaction among lactobacilli and yeast contaminants in RSM matrix. Scatter plot of the position of strains of *L. plantarum* is based on correlations among its functional properties.

### 3.2.2. The Sensitivity of Yeast Strains

As the *L. plantarum* strains showed variability in their antifungal effects, yeast contaminants showed variable sensitivity to *L. plantarum* strains and their products. Five genera of yeast (*Candida* spp., *Trichosporon* spp., *Debaryomyces* spp., *Kluyveromyces* spp., and *Geotrichum* sp.) including the unequal number of strains were tested. The sensitivity of the yeast contaminants during the test on artificial media (Figure 9) was significantly different among yeast strains (Table A4). Significant variability in yeast growth based on index scale value was noted within five yeast genera (Table S1). *Trichosporon* species shown to be the most sensitive to *L. plantarum* species. High index values were also obtained for *Candida apicola*, *C. atlantica*, *Debaryomyces subglobosus*, and *D. hansenii* strain 47. On the contrary, *C. krusei*, *C. inconspicua*, *Kluyveromyces marxianus* strains 270 and 258, and *Galactomyces candidum* strain 1061 demonstrated the maximum tolerance to *L. plantarum* strains and their products in the overlay method on artificial media.

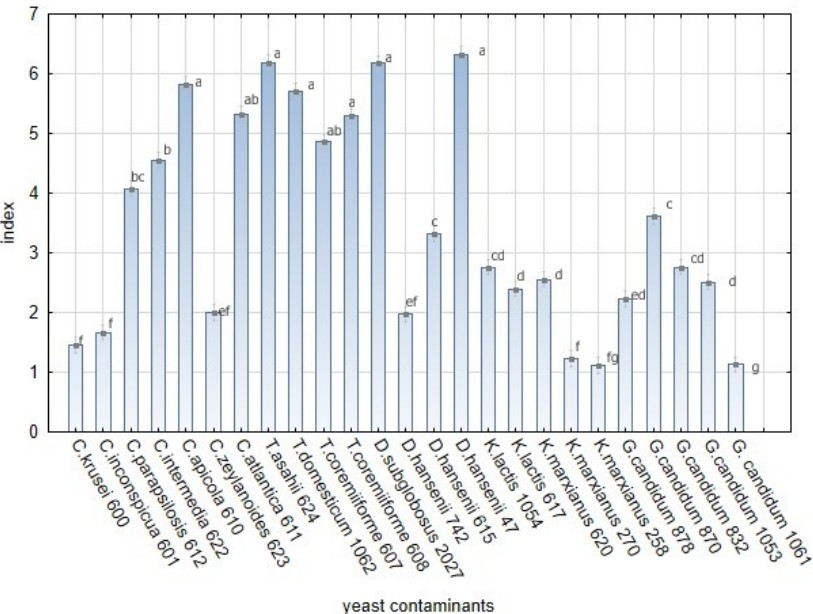

**Figure 9.** The sensitivity of yeast strains to *L. plantarum* strains in in vitro overlay tests. The bars represent the means of index value and SEM. $F_{(24, 1050)} = 400.009$; $p \leq 0.00\,E \leq 0.05$ (Factorial ANOVA, Statistica Software v. 12, Post Hoc Tukey's HSD Test). The index letter indicates significant differences. Significant at $\alpha \leq 0.05$.

The influences of the functional properties of lactobacilli on yeast genera were tested as covariate factors. Most of the tested yeast strains belonging to *Candida* spp., *Galactomyces* spp, and *Debaryomyces* spp. proved tolerant to the lactic and acetic acid content produced by lactobacilli. The bacteriocin profiles of lactobacilli significantly influenced sensitivity of almost all yeast strains except for *Kluyveromyces*. The presence of chitinase genes was found to be a significant trait only for the suppression of *Trichosporon* spp. and *Debaryomyces* spp. on RSM media. The significant variability among yeast indicates that the sensitivity of yeast to *L. plantarum* is an interspecific and intraspecific property, depending on origin of isolates and environmental factors. The overlay method based on artificial media was optimised for the growth of lactobacilli and yeast in vitro. Thus, the results are far different from those obtained in the dairy matrices. The development of yeast contaminants on RSM with lactobacilli was suppressed by up to 60% compared with controls (Figure 10). Significant variability in the average sizes of yeast colonies was noted within five yeast genera (Table S2). The functional properties of lactobacilli were tested as covariate factors. For yeast growth on media with RSM, the amount of lactic and acetic acids produced by lactobacilli were determined to be non-significant factors. The variability in the size of colonies was significant in the tested genera of yeast except for *Kluyveromyces*. The detailed information related to interactions among 20 *L. plantarum* strains and 25 yeast strains in overlay method are summarized in Supplementary Table S3. Most studies on antifungal effects have targeted filamentous fungi. The antifungal activity of lactic acid bacteria against a broad spectrum of potential yeast contaminants was tested in dairy-mimicking models in a previous study [19]. Even though the yeast and filamentous fungi belong to ascomycetous and basidiomycetous fungi there are important differences in their lifestyle. The spore germination and hyphal growth, including synthesis of the fungal cell wall, is the most sensitive stage to environmental changes. In the case of yeast, the cell wall is completed during the budding process. Thus, this step can cause difference in sensitivity of the yeast and filamentous fungi. The high sensitivity of *Candida albicans* to products of lactobacilli during the filamentous stage is declared in a previous study [44]. The interactions between yeast and *L. plantarum* are conditioned by intraspecific and interspecific characteristics of both inter-acting organisms, i.e., lactobacilli and yeast [13,31]. Yeast species such as *C.*

*apicola*, *C. atlantica*, *T. asahii*, *D. subglobosus*, and *D. hansenii* 615 showed high sensitivity to almost all strains of *L. plantarum* tested. The sensitivity of these strains significantly influenced the results. *C. apicola* and *C. atlantica* were identified in brines [54] and as part of cheese micro-flora in cheese with a long ripening period (up to 60 days) [55,56]. However, their effect on cheeses is not confirmed to be detrimental. The interaction of these two species with LAB has not yet been tested. *T. asahii* was identified in indigenous cheeses, brines, and whey [1]. In our study, *T. asahii* showed significant sensitivity to strains of *L. plantarum*. A variety of *Lactobacillus spp.* in a dairy matrix was tested by [52]. The colony size of *T. domesticum* and *T. coremiiforme* strains growing on reconstituted milk with *L. plantarum* differed from that of the control by 65%, whereas *T. asahii* differed by 93%.

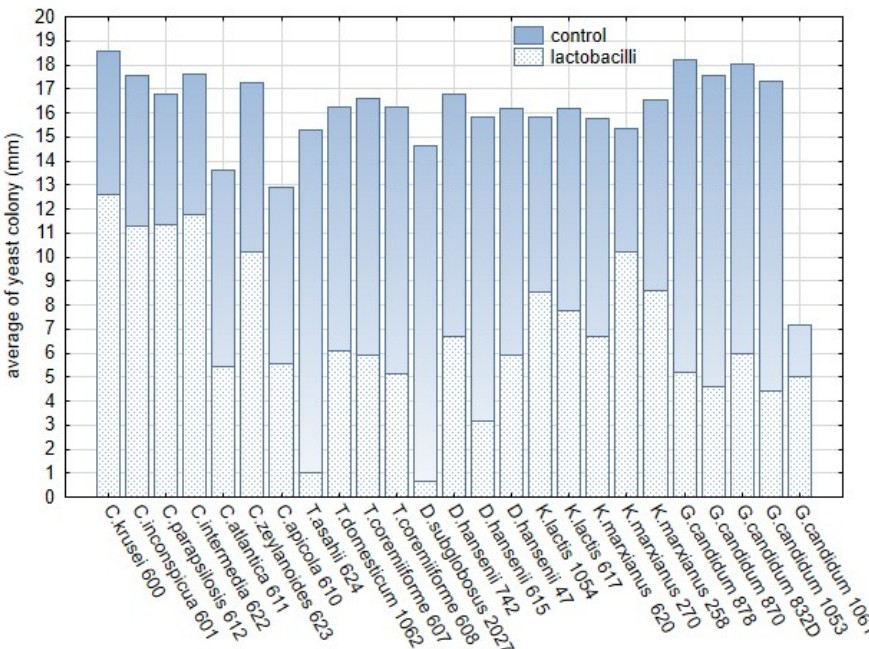

**Figure 10.** The comparison of the average sizes (mm) of the yeast colonies growing on malt extract agar supplemented with reconstituted milk with *L. plantarum* strains in and control variants without *L. plantarum*. The dotted bars represent the average of the yeast colony, blue part represents the control variant without lactobacilli.

The in vitro studies confirmed that *C. krusei*, *C. inconspicua*, *C. zeylanoides*, and strains of *K. marxianus* and *G. candidum* were the most tolerant to *L. plantarum* strains on artificial media (Table A4, Supplementary Table S3). In contrast, the inhibitory effect was apparent on RSM media, although the colony size was only about 30% less than the control. These findings reveal that a milk matrix supports the inhibitory effect of *L. plantarum*, and the inhibitory effect observed for each combination can differ from tests on artificial media [1]. The strains representing obligate pathogens, *C. krusei*, *C. parapsilosis*, and *C. inconspicua*, were included in experiments because they are frequently identified in dairy matrices. In general, these three strains were observed to be the most tolerant species to *L. plantarum*. The different by-products of *L. rhamnosus*, *L. gasseri*, *L. acidophillus*, *L. paracasei*, *L. crispatus*, and *L. jensenii* were tested to explore the biofilm adhesion and inhibition of non-albicans species in models simulating the environment of the human body [57]. Similarly, *T. asahii* is an obligate pathogen with importance in human and veterinary medicine [58]. However, *T. asahii* and other *Trichosporon* spp. are frequent in raw milk products, cheeses, and brines [1]. *T. asahii* was the most sensitive strain that did not grow on reconstituted milk with *L. plantarum* strains. In addition, the in vitro dual tests showed that *T. asahii* was highly sensitive to *L. plantarum*. The strain-specific sensitivity also occurred in *Debaryomyces* and *Kluyveromyces* spp., which represent common yeast species in the dairy environment and

products. *Geotrichum/Galactomyces* spp. were strongly suppressed by *L. plantarum* strains on RSM media when compared with artificial media

## 4. Conclusions

The screening of *L. plantarum* for functional traits followed by testing on artificial and RSM matrix enables the selection of effective strains with antifungal activity against yeast contaminants. As shown, the strains that performed the antifungal activity in synthetic media cannot display the same effect on RSM media as shown in the case of *L. plantarum* strains originating from silages and raw milk cheeses. The strain 3018 from sourdough positively provided the best antifungal effect on both the media tested. All the tested strains encode the spectrum of Class 2a bacteriocin genes with some variability in clades. In MRS medium, only chitinase, non-target peptides, and proteins were produced extracellularly. There are no existing comparative data for extracellular production of bacteriocins, chitinase, and the antifungal effect of *L. plantarum* in RSM media. The results indicate that the bacteriocin spectrum might predict the antimicrobial and antifungal ability of *L. plantarum*. The competitive resources such as silages, raw milk cheeses and sourdoughs with naturally occurring fungal organisms can provide more effective strains than the uniform environment of the milk. Although lactic acid production is the renowned trait of the antifungal effect, this study showed that its production in artificial media and RSM media is highly variable within the strains and their groups of origin. Only strain 3018 from sourdough produced significantly high amount of lactic and acetic acid in MRS and RSM medium. Thus, the final antifungal effect results from the synergistic interactions based on the enzymatic and metabolic activity of strains under specified conditions. The sensitivity and tolerance of the yeast strains also play essential roles in evaluating antifungal effects. The testing of high species richness of the yeast contaminants, including the isolates from specific dairy matrices or environments, can confirm the broad spectrum of antifungal activity of *L. plantarum* strain compared with studies employing only a few strains. Then, the strain exhibiting the antifungal activity should be tested for technological properties (salt tolerance, temperature, interactions with starter cultures, Aw) stepwise when supposed to be practically used. Although the dairy products are naturally populated with *L. plantarum*, the antifungal activity of strains originating from milk was weak and insufficient compared with strains from a fermentative environment. The application of protective strains of *L. plantarum* with adequate technological properties reflects the current demands on dairy products, including probiotic potential and microbial biocontrol against yeast spoilers. Thus, the supplementation of fermented dairy products with the suitable culture of *L. plantarum* significantly contribute to sustainable development in the field of fermented foods [59]. All in all, there is no existing uniform strain of *L. plantarum* that suppressed or eliminated all the spectrum of yeast contaminants but only the strain *L. plantarum* that perform antifungal effect against the broad spectrum of yeast contaminants under defined conditions. The identification of functional traits of *L. plantarum* strains has been shown to be an innovative and effective tool for pointed selection and understanding the interaction among lactobacilli and yeast contaminants.

**Supplementary Materials:** The following are available online at https://www.mdpi.com/article/10.3390/fermentation8010014/s1. Table S1: The factorial analysis within the yeast genera. The categorical predictors—functional properties of *L. plantarums* strains are employed as covariates. The variability in inhibition is based on logarithm of index scale value for overlay method. ANCOVA, $p \leq 0.05$, Statistica Software v. 12.1. The index letters ns means—non-significant differences. Table S2 The factorial analysis within the yeast genera. The categorical predictors—functional properties of lactobacili in milk matrice are employed as covariates. The variability in the inhibition is based on the averages of yeast colony (mm) on modified overlay method with skimmed milk. ANCOVA, $p \leq 0.05$, Statistica Software v. 12.1. The index letters ns means—non-significant differences. Table S3 The interactions among *L. plantarum* strains and yeast strains based on the logarithm of index scale value by mean, standard deviation (s.d.) and standard error of means (s.e.m.). The data resulted from Factorial ANOVA, Statistica 12v., Significant at $\alpha \leq 0.05$. Table S4 The variability of yeast colony

average (mm) characterised by mean, standard deviation (s.d.) and standard error of means (s.e.m.). The data resulted from Factorial ANOVA, Statistica 12v., Significant at $\alpha \le 0.05$.

**Author Contributions:** Conceptualization, M.K., J.C., O.B. and V.D.; methodology, M.K., J.C., O.B. and V.D.; software, J.C., M.K.; validation, J.C., M.K. and O.B.; formal analysis, M.K., O.B. and J.C.; investigation, V.D.; resources, Z.D.; data curation, J.C., M.K.; writing—original draft preparation, M.K. and J.C.; writing, M.K. and J.C.; visualization, M.K. and J.C.; supervision, M.K. and V.D.; project administration, M.K. and V.D.; funding acquisition, M.K. and V.D. All authors have read and agreed to the published version of the manuscript.

**Funding:** This research was funded by THE MINISTRY OF AGRICULTURE OF THE CZECH REPUBLIC, MZE-RO1421, THE NATIONAL AGENCY FOR AGRICULTURAL RESEARCH OF CZECH REPUBLIC, grant numbers QK1910036, QK19110024 and The National Programme on Conservation and Utilization of Plant, Animal and Microbial Genetic Resources Important for Food and Agriculture (NPGR) belonging to Ministry of Agriculture of the Czech Republic.

**Institutional Review Board Statement:** Not applicable.

**Informed Consent Statement:** Not applicable.

**Data Availability Statement:** The data that supports the findings of this study are present in Appendix A and in the Supplementary Materials.

**Conflicts of Interest:** All the authors declare that they have no relevant conflict of interest. The funders had no role in the design of the study; in the collection, analyses, or interpretation of data; in the writing of the manuscript, or in the decision to publish the results.

**Appendix A**

**Table A1.** Production of organic acids by strains of *L. plantarum* in different media. ANOVA, Statistica v. 12, Post Hoc Tukey's HSD Test. The data represents logarithmically transformed index scale values, standard deviation (SD) and standard error of means (SEM). The index letters indicate the significant differences. Significant at $\alpha \le 0.05$.

| Origin | Strain | Lactic Acid (mg.100 mL$^{-1}$) | | | Acetic Acid (mg.100 mL$^{-1}$) | | |
|---|---|---|---|---|---|---|---|
| | | **MRS** | **MRS + FGGM** | **RSM** | **MRS** | **MRS + FGGM** | **RSM** |
| Silage | 191 | 1172.94 ± 0.02 | 1387.54 ± 0.02 [b] | 959.38 ± 0.01 [b] | 484.64 ± 0.02 | 582.63 ± 0.02 [ba] | 181.49 ± 0.02 [a] |
| | 196 | 1356.38 ± 0.02 [b] | 1410.04 ± 0.05 [b] | 977.42 ± 0.10 [ab] | 484.86 ± 0.03 | 574.32 ± 0.02 [ab] | 199.88 ± 0.02 [a] |
| | 182 | 1169.07 ± 0.03 | 1431.95 ± 0.05 [a] | 1009.59 ± 0.02 [a] | 578.92 ± 0.06 [a] | 621.50 ± 0.06 [a] | 171.85 ± 0.08 [b] |
| | 185 | 1232.11 ± 0.01 | 1459.72 ± 0.02 [a] | 985.53 ± 0.12 [ab] | 568.36 ± 0.02 [a] | 533.62 ± 0.08 | 175.62 ± 0.06 [b] |
| | 187 | 1133.74 ± 0.02 | 1554.79 ± 0.03 [a] | 911.02 ± 0.06 [bc] | 523.60 ± 0.02 [ab] | 603.99 ± 0.07 [a] | 176.34 ± 0.02 [b] |
| Raw cheese | L16 | 1267.44 ± 0.05 [b] | 1381.41 ± 0.02 [b] | 1016.68 ± 0.12 [a] | 566.20 ± 0.06 [ab] | 643.32 ± 0.06 [a] | 160.88 ± 0.13 |
| | L17 | 1311.20 ± 0.05 [b] | 1292.19 ± 0.07 | 1008.47 ± 0.04 [a] | 601.05 ± 0.09 | 632.12 ± 0.09 [a] | 166.53 ± 0.10 |
| | L24 | 1185.13 ± 0.02 | 1309.29 ± 0.09 | 956.61 ± 0.03 [b] | 537.91 ± 0.12 | 593.13 ± 0.02 [a] | 136.43 ± 0.02 |
| | L32 | 928.19 ± 0.06 | 1197.13 ± 0.02 | 983.76 ± 0.07 [ab] | 513.72 ± 0.03 | 592.66 ± 0.05 [a] | 190.99 ± 0.05 [a] |
| | L12 | 1333.80 ± 0.02 [b] | 1401.14 ± 0.06 [b] | 422.56 ± 0.02 | 188.10 ± 0.05 | 373.62 ± 0.03 | 114.07 ± 0.02 |
| Milk | 381 | 1345.53 ± 0.06 [b] | 1497.48 ± 0.05 [a] | 821.42 ± 0.02 | 498.13 ± 0.06 | 516.03 ± 0.02 | 94.32 ± 0.09 |
| | 383 | 1353.22 ± 0.02 [b] | 1450.37 ± 0.02 [a] | 406.71 ± 0.02 | 503.27 ± 0.02 [b] | 409.29 ± 0.02 | 29.90 ± 0.07 |
| | 384 | 1526.51 ± 0.07 [a] | 1491.59 ± 0.04 [a] | 354.68 ± 0.02 | 563.49 ± 0.03 [a] | 482.58 ± 0.02 | 42.51 ± 0.05 |
| | 391 | 1545.06 ± 0.04 [a] | 1487.34 ± 0.03 [a] | 406.71 ± 0.05 | 537.35 ± 0.02 [ab] | 605.83 ± 0.05 [ab] | 94.32 ± 0.05 |
| Sourdoughs | K19-1 | 1047.83 ± 0.06 | 1286.92 ± 0.02 | 975.77 ± 0.06 [ab] | 590.32 ± 0.05 [a] | 613.09 ± 0.02 [a] | 135.65 ± 0.02 |
| | K19-2 | 1009.88 ± 0.02 | 1236.48 ± 0.05 | 685.02 ± 0.02 | 539.09 ± 0.09 [ab] | 611.01 ± 0.02 [a] | 162.56 ± 0.03 |
| | K19-3 | 1008.35 ± 0.07 | 1296.72 ± 0.04 | 724.97 ± 0.07 | 540.13 ± 0.07 [ab] | 550.78 ± 0.03 | 123.86 ± 0.12 |
| | K20-4 | 1116.30 ± 0.02 | 1020.33 ± 0.04 | 513.50 ± 0.06 | 540.13 ± 0.03 [ab] | 511.89 ± 0.11 | 150.60 ± 0.02 |
| | 3018 | 1367.88 ± 0.01 [b] | 1416.70 ± 0.02 [a] | 1025.10 ± 0.08 [a] | 537.56 ± 0.06 [ab] | 592.99 ± 0.12 [b] | 171.60 ± 0.02 [b] |

**Table A2.** The variability in antifungal effect of 20 strains of *L. plantarum* in in vitro tests on artificial media against 25 yeast strains. The data represent logarithmically transformed index scale values, standard deviation (s.d.) and standard error of means (s.e.m.). $F_{(24,1050)} = 96.87$; $p \leq 0.00$ E $\leq 0.05$. (Factorial ANOVA, Statistica v. 12. Post Hoc Tukey's HSD Test). The index letter indicates significant differences. Significant at $\alpha \leq 0.05$.

| *L. plantarum* Strains | Log Value | s.d. | s.e.m. |
|---|---|---|---|
| CCDM 191 | 1.07 [bc] | 0.77 | 0.09 |
| CCDM 196 | 1.16 [b] | 0.74 | 0.09 |
| CCDM 182 | 1.03 [bc] | 0.71 | 0.08 |
| CCDM 185 | 0.99 [bc] | 0.69 | 0.08 |
| CCDM 187 | 1.01 [bc] | 0.72 | 0.08 |
| L12 | 1.07 [bc] | 0.71 | 0.08 |
| L16 | 1.16 [b] | 0.72 | 0.08 |
| L17 | 1.03 [bc] | 0.79 | 0.09 |
| L24 | 1.04 [bc] | 0.76 | 0.09 |
| L32 | 0.96 [bc] | 0.79 | 0.09 |
| CCDM 381 | 0.94 [c] | 0.68 | 0.08 |
| CCDM 383 | 0.94 [c] | 0.71 | 0.08 |
| CCDM 384 | 0.86 [d] | 0.66 | 0.08 |
| CCDM 391 | 1.05 [bc] | 0.67 | 0.08 |
| CCDM 387 | 0.99 [bc] | 0.69 | 0.08 |
| K19-1 | 1.24 [b] | 0.62 | 0.07 |
| K19-2 | 1.18 [bc] | 0.61 | 0.07 |
| K19-3 | 1.23 [b] | 0.59 | 0.07 |
| K20-4 | 1.11 [bc] | 0.63 | 0.07 |
| CCDM 3018 | 1.32 [b] | 0.55 | 0.06 |
| controls | 0.00 [a] | 0 | 0 |

**Table A3.** The variability in the average size of yeast colonies (mm) including standard deviation (s.d.) and standard error of mean (s.e.m.) on RSM medium with *L. plantarum* strains, including the control variant, for all yeast strains. Factorial ANOVA, Statistica Software v. 12, $F_{(20,1530)} = 45.604$; $p \leq 0.5 \leq \alpha$. The index letter indicates significant differences. Significant at $\alpha \leq 0.05$.

| Strain of *L. plantarum* | Average (mm) | ±s.d. | ±s.e.m. |
|---|---|---|---|
| CCDM 191 | 7.56 [b] | 6.40 | 0.74 |
| CCDM 196 | 8.52 [b] | 6.49 | 0.75 |
| CCDM 182 | 5.71 [d] | 6.48 | 0.51 |
| CCDM 185 | 6.91 [bc] | 5.76 | 0.98 |
| CCDM 187 | 5.72 [d] | 4.56 | 0.53 |
| L12 | 3.61 [ef] | 2.67 | 0.42 |
| L16 | 6.89 [bd] | 4.46 | 0.52 |
| L17 | 8.18 [b] | 4.23 | 0.49 |
| L24 | 4.02 [e] | 3.45 | 0.51 |
| L32 | 4.34 [e] | 3.51 | 0.50 |

**Table A3.** *Cont.*

| Strain of *L. plantarum* | Average (mm) | ±s.d. | ±s.e.m. |
| --- | --- | --- | --- |
| CCDM 381 | 5.34 [de] | 4.51 | 0.52 |
| CCDM 383 | 5.19 [de] | 4.22 | 0.53 |
| CCDM 384 | 4.96 [de] | 3.52 | 0.60 |
| CCDM 391 | 6.50 [d] | 4.53 | 0.53 |
| CCDM 387 | 5.64 [de] | 4.24 | 0.56 |
| K19-1 | 4.99 [de] | 3.21 | 0.37 |
| K19-2 | 4.45 [e] | 3.94 | 0.46 |
| K19-3 | 3.11 [e] | 2.44 | 0.40 |
| K20-4 | 8.34 [b] | 4.78 | 0.55 |
| CCDM 3018 | 2.17 [f] | 3.61 | 0.42 |
| controls | 17.95 [a] | 2.39 | 0.27 |

**Table A4.** The variability in sensitivity 25 yeast strains to all strains of *L. plantarum*. The data represents logarithmically transformed index scale values, s.d. and s.e.m. $F_{(24, 1050)} = 400.009$; $p \leq 0.00 \, E \leq 0.05$. (Factorial ANOVA, Statistica v. 12. Post Hoc Tukey's HSD Test). The index letter indicates significant differences. Significant at $\alpha \leq 0.05$.

| Yeast Strain | Log Value | ±s.d. | ±s.e.m. |
| --- | --- | --- | --- |
| *Candida krusei 600* | 0.24 [f] | 0.46 | 0.06 |
| *C. inconspicua 601* | 0.39 [f] | 0.47 | 0.06 |
| *C. parapsilosis 612* | 1.32 [bc] | 0.45 | 0.06 |
| *C. intermedia 622* | 1.42 [b] | 0.5 | 0.06 |
| *C. apicola 620* | 1.66 [a] | 0.54 | 0.07 |
| *C. zeylanoides 623* | 0.59 [ef] | 0.46 | 0.06 |
| *C. atlantica 611* | 1.58 [ab] | 0.52 | 0.07 |
| *Trichosporon asahii 624* | 1.77 [a] | 0.41 | 0.05 |
| *T. domesticum 1062* | 1.69 [a] | 0.39 | 0.05 |
| *T. coremiiforme 607* | 1.53 [ab] | 0.38 | 0.05 |
| *T. coremiiforme 608* | 1.61 [a] | 0.4 | 0.05 |
| *Debaryomyces subglobosus 2027* | 1.75 [a] | 0.46 | 0.06 |
| *D. hansenii 742* | 0.51 [e] | 0.53 | 0.07 |
| *D. hansenii 615* | 1.12 [c] | 0.44 | 0.06 |
| *D. hansenii 47* | 1.79 [a] | 0.41 | 0.05 |
| *Kluyveromyces lactis 1054* | 0.92 [cd] | 0.47 | 0.06 |
| *K. lactis 617* | 0.82 [d] | 0.35 | 0.04 |
| *K. marxianus 620* | 0.85 [d] | 0.45 | 0.06 |
| *K. marxianus 270* | 0.15 [f] | 0.29 | 0.04 |
| *K. marxianus 258* | 0.08 [g] | 0.22 | 0.03 |
| *Geotrichum candidum 878* | 0.66 [d] | 0.53 | 0.07 |
| *G. candidum 870* | 1.19 [c] | 0.48 | 0.06 |
| *G. candidum 832* | 0.89 [cd] | 0.54 | 0.07 |
| *G. candidum 1053* | 0.83 [d] | 0.42 | 0.05 |
| *G. candidum 1061* | 0.09 [g] | 0.23 | 0.03 |

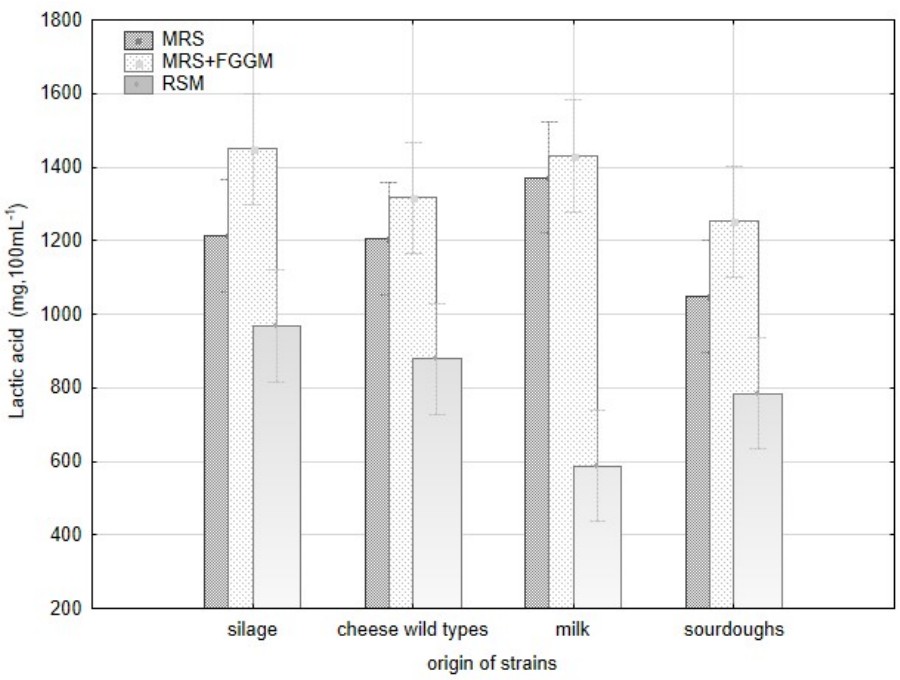

**Figure A1.** Modification of the pH value by *L. plantarum* strains with respect to strain origin and three cultivation media-MRS, MRS + FGGM and reconstituted milk.

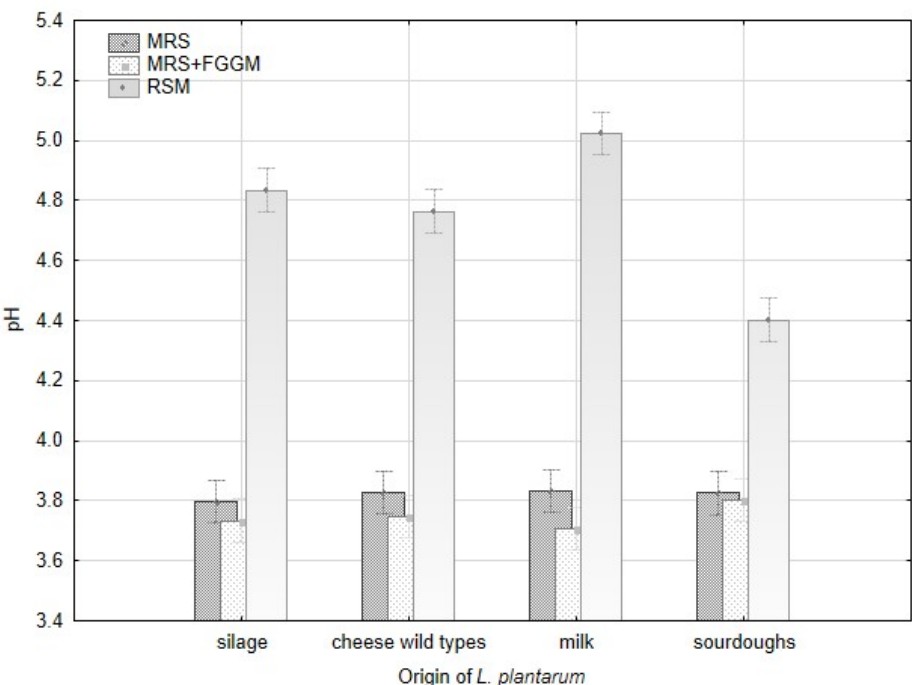

**Figure A2.** Variability in production of lactic acid by *L. plantarum* strains with respect to strain origin and three cultivation media- MRS, MRS + FGGM and reconstituted milk.

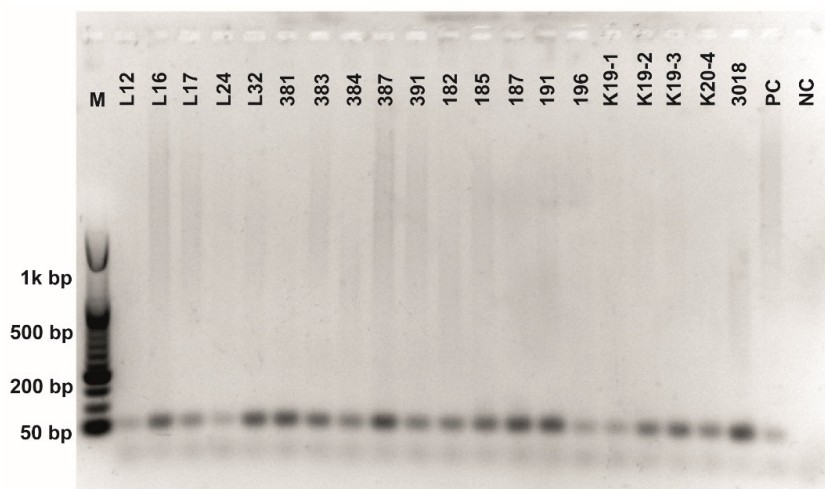

**Figure A3.** Class 2a bacteriocin gene belonging to the Clade 1 amplification. O'RangeRuler™ 50 bp DNA Ladder was used as a molecular marker. PC-positive control by strain ATTCC 14917.

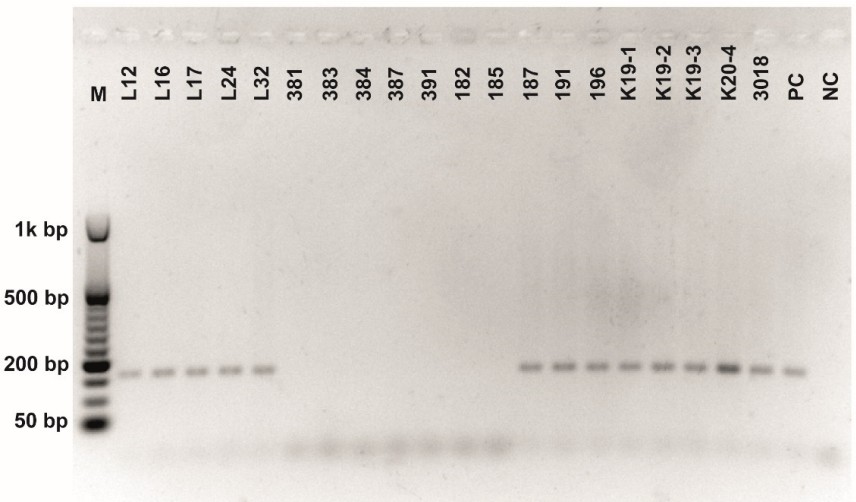

**Figure A4.** Class 2a bacteriocin gene belonging to the Clade 2 amplification. O'RangeRuler™ 50 bp DNA Ladder was used as a molecular marker. PC-positive control by strain ATTCC 14917.

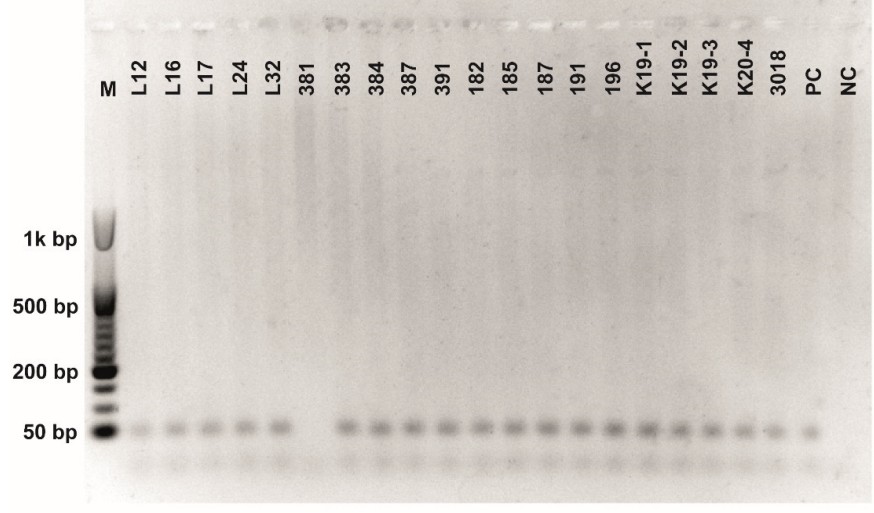

**Figure A5.** Class 2a bacteriocin gene belonging to the Clade 4 amplified using first primer pair (Table 3). O'RangeRuler™ 50 bp DNA Ladder was used as a molecular marker.

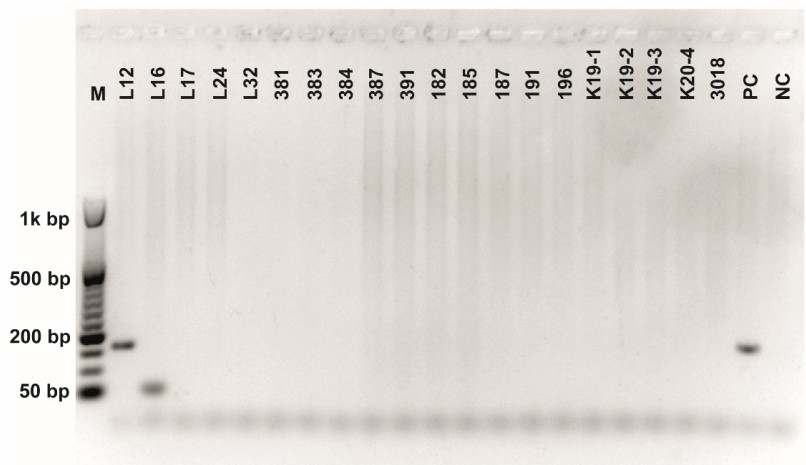

**Figure A6.** Class 2a bacteriocin gene belonging to the Clade 4 amplified using second primer pair (Table 3). O'RangeRuler™ 50 bp DNA Ladder was used as a molecular marker. PC-positive control by strain ATTCC 14917.

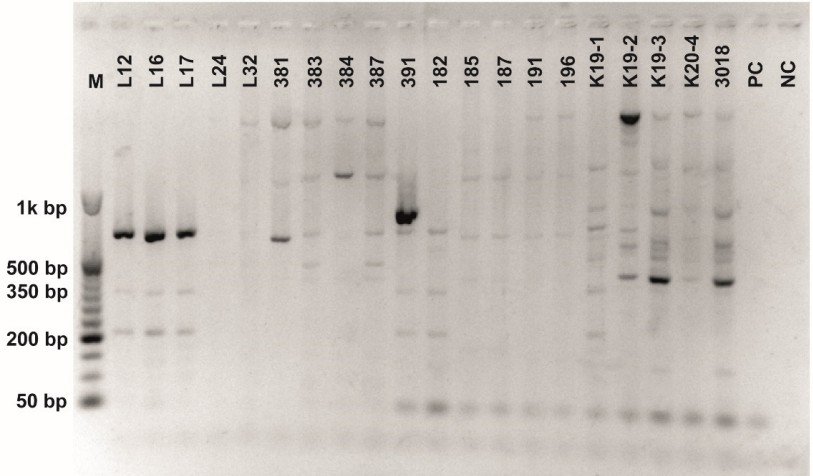

**Figure A7.** Class 2a bacteriocin gene belonging to the Clade 5 amplified using first primer pair (Table 3). O'RangeRuler™ 50 bp DNA Ladder was used as a molecular marker. PC-positive control by strain ATTCC 14917.

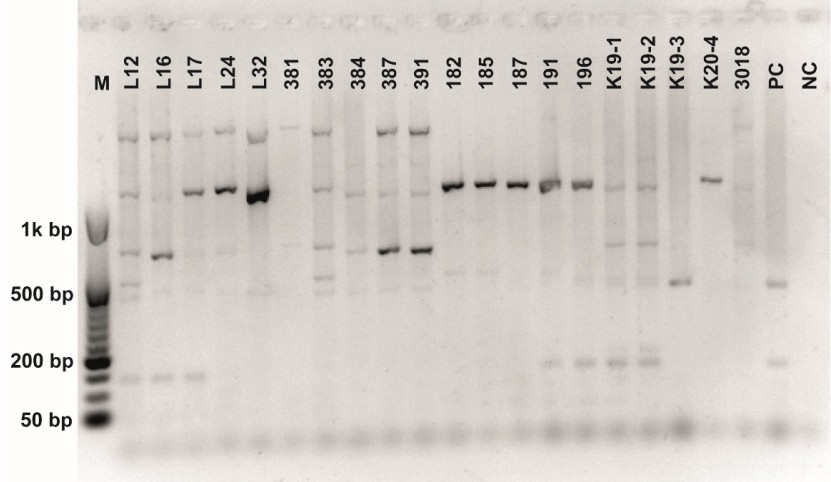

**Figure A8.** Class 2a bacteriocin gene belonging to the Clade 5 amplified using second primer pair (Table 3). O'RangeRuler™ 50 bp DNA Ladder was used as a molecular marker. PC-positive control by strain ATTCC 14917.

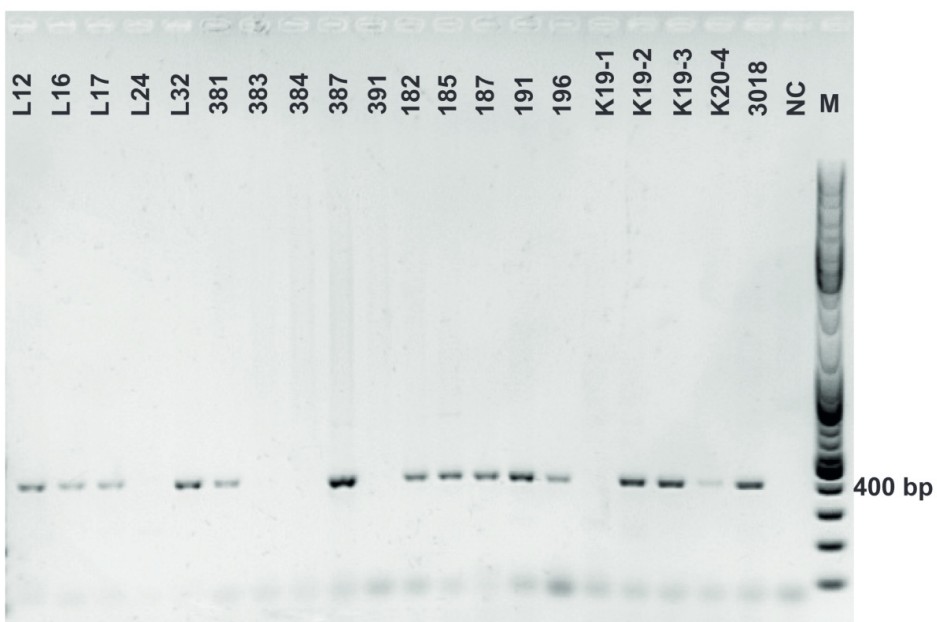

**Figure A9.** Chitinase (*chiA*) amplification. Gene encoding chitinase was detected in *L. plantarum* strains using primers chiFEMSF and chiFEMSR. GeneRuler™ DNA Ladder Mix was used as a molecular marker.

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
