# Peer review of "The Interactions among Isolates of Lactiplantibacillus plantarum and Dairy Yeast Contaminants: Towards Biocontrol Applications"

_fermentation, doi:10.3390/fermentation8010014_

Round 1

Reviewer 1 Report

In the manuscript has tested the antimicrobial capacity of 20 strains of Lactiplantibacillus plantarum versus 25 species of yeasts contaminants. In the first step, a screening of the presence of bacteriocin gene (class 2a bacteriocin and chitinase gene) and a study of physicochemical parameters on Lb. plantarum were done (genetic and chemical characterization). The second step has been an antifungal screening of lactobacilli bacteria vs yeast. How the author declares, the screening was done on artificial condition and not in food-system or real environmental place. In fact, medium of laboratory were used (MRS modify, with adjunct sugars; skim milk modified). I think this work is a preliminary study to understand the efficiency of LABs vs yeasts because the author has used medium of laboratory and different Lb. plantarum isolated from different sources. The data are interesting, although sometimes some parts are not very clear. It is my opinion reconsider the manuscript after major revision

Some questions:

Could the author add some information on introduction section of the 2a bacteriocins and chitinase gene? Why are they so important for this study? Please, add some information and references.

Why did the author talk of antifungal activity against yeasts and not antimicrobial action? Please explain better this part.

Experimental set-up: the author revitalised the lyophilised cultures in 16% reconstituted milk and after he/she cultivated in MRS modified (with sugar) to increase the production of bioactive products. My question is: why did the author use milk to revitalise? The strain of L. plantarum were isolated by different source (sourdough or silage). Another question: did MRS modified create the real condition (for example food environment, especially milk/cheese system)? Why is it used this medium?

Although, the author measured the physicochemical parameters with laboratory condition: why? I think that a good contribution to the community scientific is create a real condition. For example, food system (skim milk, i.e.). Could the author think to create a food system model to reinforce these data collected?

Please, the author would control whole manuscript: L. plantarum is not often in italic mood and after the first time that he/she write the whole name, he/she could write in acronym mood (L. plantarum). Often the name has written full (i.e. P7, L233).

Minor comment:

P2, L97: please explain acronym FGGM;

P6, L182-189: the ref. num 27, a review, I have not found this method. Please, add a specific reference;

P8, L268-279: the data on fig A7 and A8 cannot help to understand the presence of clade 5-1 and 5-2 gene. A lot of samples (but almost all) have multiple bands (sometimes very low intensity band). How the author can talk of the presence of bacteriocin genes belonging to clade 5?

P9, L287: Lc. is necessary to write in whole form, because it is the first time that the author cite this name strain, please correct;

Table 1 and 2 are not cited in the correct section. Please, the author could add Table 1 in the par. 2.1 and Table 2 in the par 2.2.

Figure 1: what mean columns number? Please explain;

Figure A1 and A2: I think is better write RSM and not skim milk, or did the author use another medium? Please correct.

Figure from A3 and A7: please add information of size band and explain acronym NC on caption;

Table 5: Could the author explain the factors trend? What are groups F1 or F2, for example?

Author Response

Dear Sir/Madam, 

Thank you for the detailed revision of our manuscript and adequate recommendations. We did our best to improve the quality of the manuscript to fill your requests. All changes are marked in the manuscript. I have split the result and discussion into one section because of the many topics discussed. Also, for a better explanation of organic acid production, I have added Table A1. I believe that the changes made in the manuscript increased its quality.

Sincerely 

Dr. Miloslava Kavková

Reviewer 2 Report

Congratulations to the researchers. There is a lot of work and it has an approach that is a very good starting point for new and interesting research and applications.

However, I have to say that the text is not so careful and does not accompany the effort made.

The homogeneity of the text, the quality of the figures and the nomenclature used should be improved.

Everything is perfectly solvable but I find myself in the situation of indicating to them that an intense work of revision and edition of the text is required.

Also, I miss a greater bibliographic review work in the part of the results (You have handled a wide bibliography, however in some cases you use up to 3 references for the same phrase and, nevertheless, in the results you have used very few references).

I put some examples below. Although, I strongly urge you not to just stick with those I quote, check all the text better.

Line 17: change C lass IIa bacteriocin to class IIa bacteriocin

Line 64: Same nomenclature in all text: (better in roman number)

Line 123 and 139: GENE GENIUS. In some articles I have seen it in a single word and in others separately, but I think the whole word should not be capitalized: Gene Genius or GeneGenius.

Check all trade names, some of them were written in capital letters. For example, MERCK better Merck.

Line 180: The results are presented as mg per 100 mL (SI better: mg 100 mL−1)

Lines 182, 186, 188, 192, 194, 199, 203: L. plantarum on italic, please (L. plantarum). Check all the text.

Line 189: lactobacillus must be change (Lactobacillus or lactobacilli).

Line 215: Post Hoc Tukey’s HSD Test. I think it is better post hoc Tukey’s HSD test

Lines 216-217:  You have defined the following categories: total inhibition, inhibition, and no inhibition. I think the second (inhibition) should be partial or intermediate inhibition (as you have specified correctly in table 4).

Line 231: Three levels of inhibition represent the level of inhibition for PCA analysis. It is redundant. Rewrite the sentence.

Line 280: Class 2a better class IIa.

Figure 2: bars with points were defined inhibition 2-5, I think it should be partial inhibition.

The quality of all the figures are really bad. In addition, the texts that present the figures are read blurry and giving a feeling of poor quality that does not correspond to the work done.

Check all the text. There are a lot of common names that start with capital letters in the middle of sentences or after commas. A clear example would be the footnote of figure 1.

In Figure 4, in the principal component analysis, inhibition (Inh) appears again instead of partial inhibition.

In figure 6 the color of the bars corresponding to cheese and sourdough are almost the same. Fill with horizontal lines or another that is more easily distinguishable.

In addition, there is no homogeneity with the filling of the bars in figure 3. In figure 6 the bars with grid correspond to milk and in figure 3 to sourdough. It is better that there is homogeneity throughout the document.

Line 404: You wrote “The strains with partial inhibitory effects are

placed in the upper right part of the PCA scatter plot.”. But in figure 4 you can see how in the upper right corner you can read no inhibition. Partial inhibition is correlated with the negative part of PC 1.

Regarding the conclusions, in view of the results obtained, I was eager to read them.

I have to say that the first part is not conclusions as such, they are a summary of the results.

I think that, in view of the results and the discussion they carry out, the conclusions are not sufficiently consistent with them.

While it is true that other tests can give more information and will be necessary, it seems that the conclusions are very "tepid" considering the results. I would give the conclusions a different point of view.

Author Response

Dear Sir/Madam, 

Thank you for the detailed revision of our manuscript and adequate recommendations. We did our best to improve the quality of the manuscript to fill your requests. All changes are marked in the manuscript. I have split the result and discussion into one section because the many topics discussed and relevant discussion to the results seem to be clearer. We have improved the inconsistency in figures 3 and 6 and commentary of PCA analysis (figure 4). We added one more table (A1) to the appendix to clarify the results on organic acid production. The study remains the innovative approach to the evaluation of L. plantarum strains. There are not exist too many available and relevant resources for discussion on the field of L. plantarum and for such a wide spectrum of the yeast tested. We completely change the conclusions to summarize all the aspects of the study. The figures with PCR products were improved in quality. I hope that the changes made in the manuscript increased its quality.

Sincerely

Dr. Miloslava Kavková
